# Polysulfur-based bulking of dynamin-related protein 1 prevents ischemic sulfide catabolism and heart failure in mice

Akiyuki Nishimura [1,2,3], Seiryo Ogata [4], Xiaokang Tang [1,2,3], Kowit Hengphasatporn[5], Keitaro Umezawa [6], Makoto Sanbo [1], Masumi Hirabayashi [1], Yuri Kato[7], Yuko Ibuki [8], Yoshito Kumagai[7], Kenta Kobayashi[1], Yasunari Kanda [9], Yasuteru Urano [10,11], Yasuteru Shigeta [5], Takaaki Akaike [4] & Motohiro Nishida [1,2,3,7] ✉

The presence of redox-active molecules containing catenated sulfur atoms (supersulfides) in living organisms has led to a review of the concepts of redox biology and its translational strategy. Glutathione (GSH) is the body's primary detoxifier and antioxidant, and its oxidized form (GSSG) has been considered as a marker of oxidative status. However, we report that GSSG, but not reduced GSH, prevents ischemic supersulfide catabolism-associated heart failure in male mice by electrophilic modification of dynamin-related protein (Drp1). In healthy exercised hearts, the redox-sensitive Cys644 of Drp1 is highly S-glutathionylated. Nearly 40% of Cys644 is normally polysulfidated, which is a preferential target for GSSG-mediated S-glutathionylation. Cys644 S-glutathionylation is resistant to Drp1 depolysulfidation-dependent mitochondrial hyperfission and myocardial dysfunction caused by hypoxic stress. MD simulation of Drp1 structure and site-directed mutagenetic analysis reveal a functional interaction between Cys644 and a critical phosphorylation site Ser637, through Glu640. Bulky modification at Cys644 via polysulfidation or S-glutathionylation reduces Drp1 activity by disrupting Ser637-Glu640-Cys644 interaction. Disruption of Cys644 S-glutathionylation nullifies the cardioprotective effect of GSSG against heart failure after myocardial infarction. Our findings suggest a therapeutic potential of supersulfide-based Cys bulking on Drp1 for ischemic heart disease.

Recent changes in therapeutic strategies for cardiac diseases have led to a broad acceptance of the concept of heart failure as a metabolic disease. Since mitochondrial dysfunction is a key feature of the development of various types of heart failure, mitochondria are expected as a potent target for heart failure therapy[1,2]. Mitochondrial quality is precisely controlled by the following GTPases: mitochondrial fission factors such as dynamin-related protein1 (Drp1) and fusion factors such as mitofusin 1/2 (Mfn1/2) and optic atrophy 1 (Opa1), and

genetic ablations of any of the GTPases reportedly cause heart failure[3–5]. We previously found that Drp1 mediates mitochondrial hyperfission-associated myocardial early senescence after myocardial infarction (MI), leading to chronic heart failure[6]. Hypoxic stress induces Drp1 interaction with actin-binding protein filamin A (FLNa), a guanine nucleotide exchange factor for Drp1, and this complex formation at the mitochondrial fission site leads to Drp1-dependent mitochondrial fission[6].

The primary role of myocardial mitochondria is to sustain high energy production while maintaining intracellular redox balance[7]. Oxidative stress due to the accumulation of reactive oxygen species (ROS) and ROS-derived electrophiles is believed to exacerbate the prognosis of ischemic heart diseases. To date, various clinical trials of antioxidant therapies including glutathione (GSH), an endogenous nucleophile that is abundantly present in living organisms, have been performed to improve outcomes in heart failure patients, but most of them are failed[8–10]. These results suggest the need to review the concept of redox-based therapy. Sulfur belongs to the group 16 elements (chalcogen), which has potent electron transfer properties in living organisms[11]. Supersulfides such as cysteine (Cys) per(poly)sulfide (CysS$_{(n)}$SH, n ≥ 1) have highly reactive catenated sulfur structures and are abundantly found in mammals[12,13]. Importantly, individual sulfur residues of supersulfides possess both nucleophilic and electrophilic characters, and supersulfides show distinctly different biological properties compared to classical sulfur metabolites[11,13,14]. We previously found that intracellular supersulfides are converted to hydrogen sulfides (H$_2$S/HS$^-$) in cardiomyocytes under hypoxia and this supersulfide catabolism is correlated with cardiac dysfunction[15]. Catenated sulfur is also incorporated into Cys residues in proteins. Protein polysulfidation and depolysulfidation cycles are post-translationally regulated by cellular redox status, and polysulfidation-based regulation of protein function has been reported in several proteins[16,17]. We previously identified that polysulfidation of rat Drp1 at Cys624 (Cys644 of human Drp1) negatively regulates its activity[13], and supersulfide depletion induces Drp1 depolysulfidation, leading to mitochondrial fission-associated myocardial senescence and fragility through the increased Drp1-FLNa complex formation[18]. This evidence suggests that Drp1 acts as a redox sensor protein that links mitochondrial quality and cellular and extracellular redox status.

GSH is the most abundant endogenous antioxidant in the body[19,20] predominantly exists as a reduced form but is converted into an oxidized form (GSSG) in response to redox status. GSSG has been considered a marker of oxidative status but leads to reversible but stable modification (S-glutathionylation) of Cys thiol on protein[21]. S-glutathionylation not only protects proteins from irreversible oxidative modification but also directly alters protein structure and function to protect cardiovascular tissues against oxidative stress[21–23]. GSSG can catalyze S-glutathionylation and release nucleophilic GSH. Here we demonstrate that GSSG administration improves cardiac dysfunction after MI in mice, whereas reduced GSH is insufficient for cardiac protection. GSSG but not GSH preferentially induced Drp1 S-glutathionylation at Cys644, which protects cardiomyocytes from hypoxic and pseudohypoxic stress-induced supersulfide catabolism including Drp1 depolysulfidation at Cys644. GSSG treatment prevented Drp1 depolysulfidation-mediated mitochondrial hyperfission and myocardial senescence. These findings prove the breakthrough therapeutic potential of GSSG for ischemic chronic heart failure.

## Results

### Oxidized glutathione mediates Drp1 S-glutathionylation at Cys644

Drp1 signaling in the regulation of mitochondrial dynamics is important for exercise performance[24,25] and stress tolerance of the heart[5,18,26]. Drp1 is a redox-sensitive protein and various redox-related post-translational modifications of Drp1 at Cys644 have been reported[13,18,27–30]. As physical exercise induces changes in glutathione redox state in various species, including humans[31,32] and mice[33], we speculated S-glutathionylation of Drp1 in exercise mice. To investigate Drp1 S-glutathionylation, hearts were isolated from sedentary and voluntary exercise mice, and S-glutathionylated proteins were precipitated by immunoprecipitation using an anti-S-glutathione antibody. Drp1 S-glutathionylation was increased in exercise mouse hearts (Fig. 1A). GSSG concentration and GSSG/GSH ratio were increased in the heart of exercise mice compared with sedentary mice, indicating the correlation between Drp1 S-glutathionylation and cellular GSSG contents (Fig. 1B and C). Next, to analyze which forms of glutathione lead to Drp1 S-glutathionylation, purified Drp1 proteins (Supplementary Fig. 1) from *E. coli.* or HeLa cells were reacted with GSH, GSSG, and S-nitrosoglutathione (GSNO), respectively. Electrophilic forms of glutathione (GSSG and GSNO) but not GSH led Drp1 S-glutathionylation (Fig. 1D). To identify the S-glutathionylation site of Drp1, redox-sensitive Cys644 was mutated. Drp1 C644W mutant showed approximately 40% less GSSG-mediated S-glutathionylation compared with wild-type (Fig. 1E). We previously identified Drp1 polysulfidation at Cys644 using a tag-switch-tag assay and the mutational analysis[18]. The polysulfidation of Drp1 Cys644 was directly detected by mass spectrometry analysis (Fig. 1F and Supplementary Fig. 2). Nearly 40% of Cys644 of Drp1-FLAG from HeLa cells was polysulfidated (CysS-SH or CysS-SSH) in basal state, whereas we could not detect basal S-glutathionylation (CysS-SG) and S-nitrosylation (CysS-NO) (Fig. 1G). S-glutathionylation of Drp1 Cys644 was detected from GSSG-treated Drp1 proteins by mass spectrometry analysis (Fig. 1H). Among all 9 cysteine residues in Drp1, Cys644 and Cys470 were most effectively S-glutathionylated by GSSG, and about half of Cys644 was S-glutathionylated by GSSG treatment (Fig. 1I). This mass spectrometry result is consistent with the finding that S-glutathionylation of Drp1 C644W mutant is reduced by approximately 40% compared to Drp1 WT (Fig. 1E). To analyze the effect of polysulfidated Cys on S-glutathionylation efficiency by GSSG, we prepared depolysulfidated or polysulfidated Drp1 by mixing His-Drp1 without or with Na$_2$S$_4$, and then treated with GSSG. We purified His-Drp1 from *E. coli.* without the protection of the polysulfide group. Basal polysulfidation levels of all cysteines were low in this sample, and Na$_2$S$_4$ treatment highly increased polysulfidation at Cys644 compared with other cysteines (Fig. 1J and Supplementary Fig. 3A and B). Compared with a high dose (3 mM) of GSSG treatment (Fig.1I), Cys470-SH, Cys644-SH and other cysteines showed a similar and low efficiency for S-glutathionylation by a low dose (1 mM) of GSSG (Supplementary Fig. 3C). Moreover, polysulfidated Cys644 by Na$_2$S$_4$ showed a significant increase in S-glutathionylation by GSSG compared with depolysulfidated Cys644 and other cysteines (Fig. 1J and Supplementary Fig. 3C), suggesting that polysulfidated Cys644 is preferentially S-glutathionylated by GSSG.

GSH predominantly existed in neonatal rat cardiomyocytes (NRCMs) and GSSG but not GSH treatment altered the intracellular GSSG/GSH ratio (Fig. 1K), indicating GSSG incorporation into cells. To analyze Drp1 S-glutathionylation in cardiomyocytes, proximity ligation assay (PLA) using anti-Drp1 and anti-S-glutathione antibodies was performed. PLA positive dots were increased in NRCMs treated with GSSG (Fig. 1L), suggesting GSSG-induced Drp1 S-glutathionylation in cardiomyocytes.

### GSSG treatment prevents mitochondrial hyperfission-associated myocardial senescence

Polysulfidated Cys is bulkier than normal Cys, and Drp1 polysulfidation at Cys644 negatively regulates its activity[13]. We speculated that the addition of bulky glutathione at Cys644 of Drp1 alters its activity. As previously reported[6], hypoxic stress triggered mitochondrial hyperfission-associated myocardial senescence. GSSG treatment suppressed mitochondrial hyperfission (Fig. 2A). Electron microscopic analysis revealed that the disruption of mitochondrial cristae structure under hypoxia was recovered by GSSG (Fig. 2B). Consistent with mitochondrial morphology, GSSG suppressed the decrease in mitochondrial membrane potential and oxygen consumption rate (OCR) by hypoxia or hypoxia/reoxygenation (H/R), respectively (Fig. 2C, D).

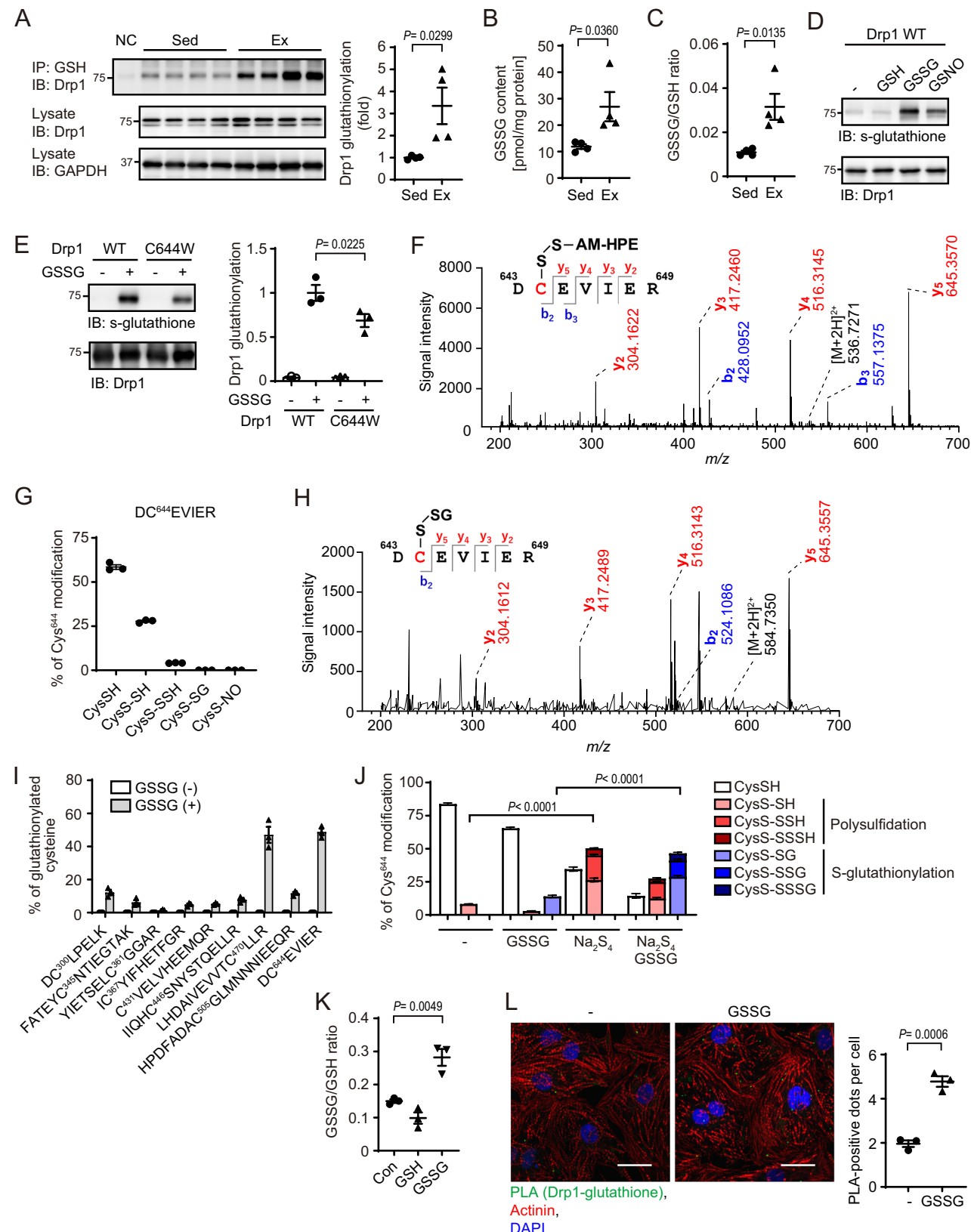

GSSG also reduced the SA-β-gal or p53 positive cellular senescence of NRCMs and improved the disruption of contractile activity after H/R (Fig. 2E, F and Supplementary Fig. 4). Additionally, to investigate the generality of cardioprotective effects of GSSG, we used human iPS-derived cardiomyocytes (hiPS-CMs). GSSG treatment improved hypoxia-induced mitochondrial fragmentation and bioenergetics and H/R-induced disruption of contractile activity of hiPS-CMs (Supplementary Fig. 5A–C). These results suggest mitochondrial- and cardioprotective roles of GSSG in hypoxic rodent and human cardiomyocytes.

**Fig. 1 | S-glutathionylation of Drp1 Cys644 by electrophilic glutathione. A** Drp1 S-glutathionylation in the hearts from sedentary mice (Sed) or voluntary exercise mice (Ex). S-glutathionylated Drp1 was immunoprecipitated and quantified. ($n = 4$ mice per condition). NC indicates immunoprecipitation without antibody as a negative control. **B, C** Quantification of total GSSG content **B** and GSSG/GSH ratio **C** in the hearts. ($n = 4$ mice per condition). **D** In vitro S-glutathionylation of Drp1. Recombinant His-Drp1 was reacted with GSH, GSSG or GSNO. ($n = 3$ independent experiments). **E** S-glutathionylation of Drp1 WT and C644W mutant by GSSG. ($n = 3$ independent experiments). **F** MS/MS spectrum for identifying polysulfidation of Cys644 in Drp1. Drp1-FLAG was purified from HeLa cells. The identified fragment ions shown in the spectra and the peptide sequence (b and y ions) evidenced the presence of persulfidated Cys labeled with HPE-IAM (marked in red). **G** Basal modification status of Cys644 in Drp1. The proportion of CysSH, CysS-SH (persulfidation), CysS-SSH (trisulfidation), CysS-SG (S-glutathionylation) and CysS-NO (S-nitrosylation) at Cys644 of Drp1-FLAG isolated from HeLa cells was quantified by mass spectrometry analysis. ($n = 3$ independent experiments). **H** MS/MS spectra for identifying S-glutathionylation of Cys644 in Drp1. Drp1-FLAG was treated with GSSG. **I** Quantification of S-glutathionylation of each cysteine residue in Drp1. Drp1-FLAG was treated with or without GSSG (3 mM). Analyzed peptide fragments containing cysteine residue are shown. ($n = 3$ independent experiments). **J** Quantification of polysulfidation and S-glutathionylation of Cys644 in Drp1. Depolysulfidated His-Drp1 was reacted with $Na_2S_4$ to prepare highly polysulfidated Drp1, and then treated with or without GSSG (1 mM). The proportion of CysSH, polysulfidation (CysS-SH, CysS-SSH and CysS-SSSH) and S-glutathionylation (CysS-SG, CysS-SSG and CysS-SSSG) at Cys644 was quantified. ($n = 3$ independent experiments). **K** Quantification of GSSG/GSH ratio in NRCMs treated with GSSG or GSH. ($n = 3$ independent experiments). **L** PLA assay using anti-Drp1 and anti-S-glutathione antibodies. PLA signals (green), counterstained with actinin (red) and DAPI (blue) were quantified. ($n = 3$ independent experiments). Scale bar, 20 μm. Data are shown as the means ± SEM. Significance was determined using two-sided unpaired t-test **A–C,L**; one-way ANOVA followed by Tukey's post-hoc test **E, J, K**. Source data are provided as a Source Data file.

## S-glutathionylation preferentially inhibits supersulfide catabolism-mediated Drp1 activation

To compare the inhibitory feature of Drp1 by polysulfidation and S-glutathionylation, Supersulfide donors ($Na_2S_3$, NaHS) or GSSG were pretreated and washed out before the incubation under hypoxia (1% $O_2$) (Fig. 3A). Hypoxia-induced activation of Drp1 was prevented by the pretreatment of GSSG but not $Na_2S_3$ and NaHS (Fig. 3B and Supplementary Fig. 6A). In addition, 30 min pretreatment of GSSG was sufficient to inhibit Drp1 activation by 1 h hypoxia, suggesting a direct effect of GSSG for Drp1 inhibition (Supplementary Fig. 6B). Depolysulfidation mimic Drp1 C644S mutant has a higher basal activity than Drp1 wild-type[18]. GSSG had no effect on the activity of Drp1 C644S, indicating the importance of Cys644 on the inhibitory effect of GSSG (Fig. 3C). GTP-bound active Drp1 gathers at the mitochondrial fission site and forms punctate structures for mitochondrial fission. Consistent with GTP-binding ability of Drp1, GSSG treatment suppressed the particle formation of GFP-Drp1 but not GFP-Drp1 C644S under hypoxia (Fig. 3D). Glutaredoxin (Grx) that are a family of GSH-dependent thiol-disulfide oxidoreductase, mainly catalyzes protein deglutathionylation. Overexpression of Grx1 partially canceled the inhibitory effect of GSSG on hypoxia-induced Drp1 particle formation and activation (Fig. 3D and E). Consistent with the result of in vitro Drp1 S-glutathionylation by GSSG but not GSH (Fig. 1D), GSH treatment did not suppress hypoxia-induced Drp1 activation and mitochondrial hyperfission (Supplementary Fig. 7A and B). Because a high dose of GSSG induced S-glutathionylation at Cys470 (Fig. 1I), the involvement of Cys470 was analyzed. Drp1 C470S mutant showed low GTP-binding and multimer formation activity in the basal condition, and hypoxia-induced activation of Drp1 C470S is drastically inhibited by GSSG treatment (Supplementary Fig. 8A and B). Moreover, co-expression of Grx1 did not alter the GTP-binding activity of Drp1 C644S with or without GSSG (Supplementary Fig. 8C), excluding the possibility that S-glutathionylation of cysteines other than Cys644 contributes to Drp1 inhibition. These results suggest that GSSG inhibits hypoxia-induced Drp1 activation through Cys644 S-glutathionylation.

Because supersulfides can receive electrons in the mitochondrial electron transport chain (ETC) system as well as oxygen[13], sulfide metabolism is closely related to mitochondrial activity. Supersulfides and $H_2S$ imaging using QS10[34] and SF7-AM[35] showed that hypoxia decreases intracellular supersulfides and increases $H_2S$ in cardiomyocytes (Supplementary Fig. 9A,B) as previously reported[15]. QS10 but not SF7-AM probe specifically responded to the administration of supersulfide donor $Na_2S_2$ (Supplementary Fig. 9C). To examine the relationship between mitochondrial activity and sulfide metabolism, simultaneous supersulfides and $H_2S$ imaging was performed in cells treatment with mitochondrial uncoupler FCCP. FCCP treatment decreased supersulfides and increased $H_2S$ in a concentration and time-dependent manner (Supplementary Fig. 9D–H), suggesting a negative correlation between supersulfides and $H_2S$ depending on mitochondrial activity. According to previous reports that electrons from ETC mediate CysSSH reduction to form $HS^-$[13] and hypoxia increases $H_2S$ formation in brain[36], these results suggest that mitochondrial depolarization under hypoxia promotes supersulfide catabolism to $HS^-/H_2S$[15]. Consistent with the changes in supersulfide catabolism, Drp1 polysulfidation was decreased under hypoxia (Fig. 3F). On the other hand, Drp1 S-glutathionylation by GSSG was sustained after additional incubation under normoxia and hypoxia (Fig. 3G). Drp1 S-glutathionylation would be more stable than polysulfidation.

## Environmental electrophiles facilitate supersulfide catabolism in cardiomyocytes, leading to Drp1-mediated cellular senescence, which is improved by GSSG

Various environmental factors may raise the risk of heart disease. We previously identified that environmental electrophile MeHg induces Drp1 activation through depolysulfidation of Cys644 and increases cardiac vulnerability to hemodynamic overload[18]. To evaluate the importance and generality of sulfur redox status (in particular polysulfidation of Drp1) for cardiac homeostasis, we investigated whether another environmental pollutant, cigarette sidestream smoke (CSS) also induces cardiomyocyte dysfunction through Drp1 redox modification. We first analyzed the effect of CSS exposure on sulfide metabolism in cardiomyocytes. The intracellular supersulfides and $H_2S$ were measured using SSip-1[37] and SF7-AM[35] probes, respectively. CSS exposure decreased supersulfides and increased $H_2S$ production, respectively (Supplementary Fig. 10A,B). CSS exposure induced similar changes in supersulfide catabolism with hypoxic stress (Supplementary Fig. 9A,B). As previously reported[38], CSS exposure induced the α subunit of the hypoxia-inducible transcription factor (HIF-1α), reflecting pseudohypoxia. Consistent with supersulfide catabolism, Drp1 polysulfidation was decreased by CSS (Supplementary Fig. 10C).

Exposure of NRCMs to low-dose CSS induced mitochondrial hyperfission and the decrease of mitochondrial membrane potential which is shown as a reduced JC-1 red/green ratio (Supplementary Fig. 10D,E). Consistent with the depolarization of mitochondrial membrane potential, CSS treatment decreased maximum respiration (Supplementary Fig. 10F). CSS exposure also induced SA-β-gal positive cellular senescence and reduced contractility of NRCMs (Supplementary Fig. 10G,H). GSSG treatment improved these dysfunctions induced by pseudohypoxic stress (Supplementary Fig. 10D–H). We previously identified that exposure of NRCMs to low-dose of MeHg leads to mitochondrial dysfunction through depolysulfidation of Drp1 at Cys644[18]. GSSG treatment also suppressed MeHg-induced mitochondrial dysfunction (Supplementary Fig. 10I). These results suggest that electrophilic bulky S-glutathionylation of Drp1 at Cys644 negatively regulates its activity and protects mitochondrial function from

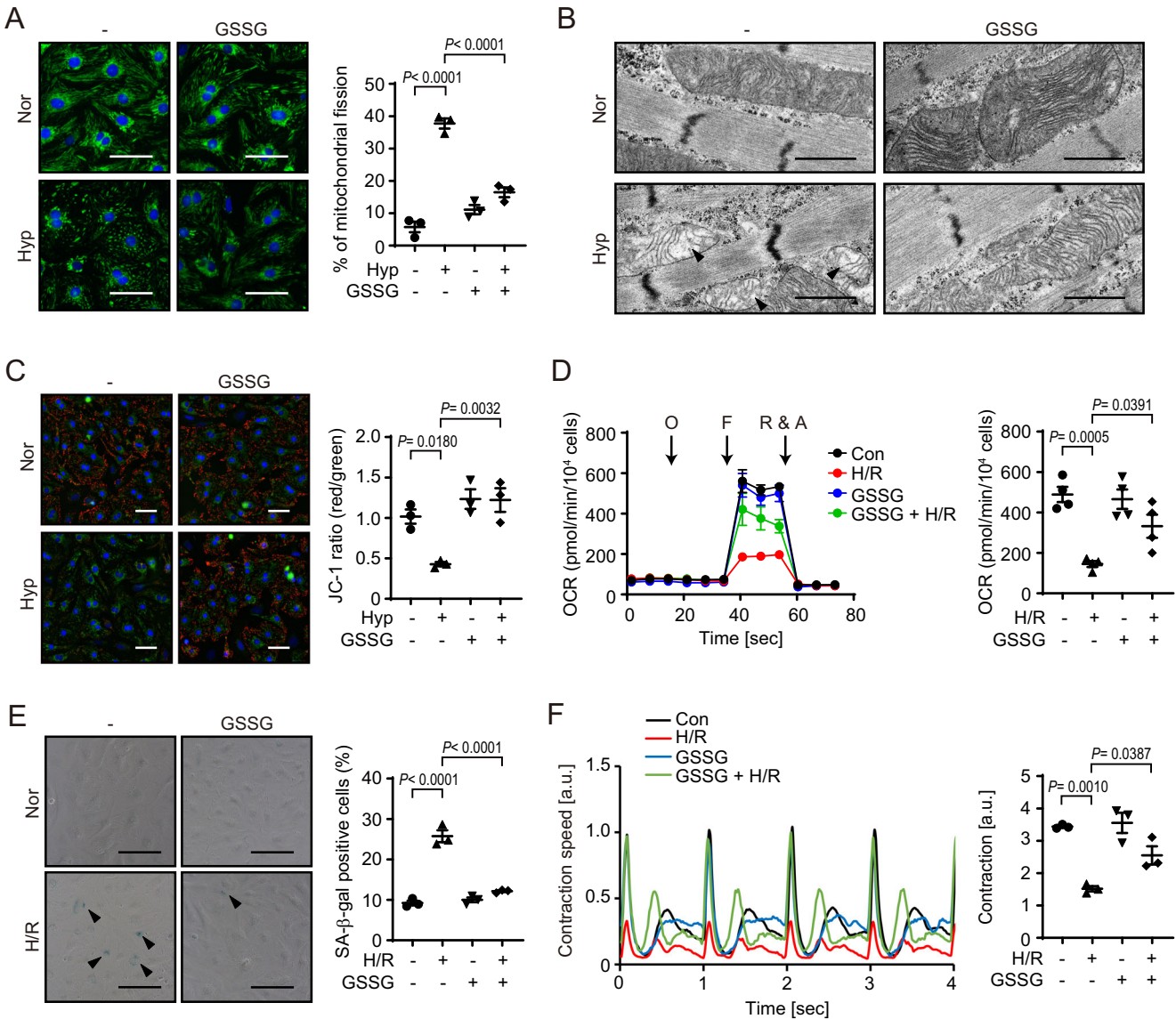

**Fig. 2 | GSSG treatment prevents hypoxia-induced myocardial dysfunction.**
**A**, **B** Representative images of mitochondrial morphology (green) from fluorescence microscopy **A** and electron microscopy **B** in NRCMs pretreated with GSSG under normoxia (Nor) or hypoxia (Hyp). The percentage of cells with vesicle-type mitochondria **A** was quantified. (*n* = 3 independent experiments). Arrowheads show mitochondria with broken cristae structure. Scale bars, 50 μm **A** or 1 μm **B**. **C** Representative images of mitochondrial membrane potential. The percentage of average JC-1 red/green ratio was quantified. (*n* = 3 independent experiments). Scale bars, 50 μm. **D** Oxygen consumption rate (OCR) in NRCMs pretreated with GSSG under Nor or hypoxia/reoxygenation (H/R). Oligomycin (O), FCCP (F), and rotenone + antimycin A (R & A) were added at the indicated timing. Right graphs show the quantitative analysis of maximal respiration. (*n* = 4 independent experiments). **E** Representative images of SA-β-gal staining in NRCMs pretreated with GSSG under Nor or H/R. The percentage of SA-β-gal positive cells (indicated by arrowheads) was quantified. (*n* = 3 independent experiments). Scale bars, 50 μm. **F** Representative traces of contraction speed of NRCMs. The contraction was calculated (*n* = 3 independent experiments). Data are shown as the means ± SEM. Significance was determined by one-way ANOVA followed by Tukey's post-hoc test. Source data are provided as a Source Data file.

supersulfide catabolism induced by not only hypoxia but also electrophilic environmental chemicals.

## GSSG administration improves cardiac function after MI
We analyzed supersulfide catabolism in an ischemic heart using chemical probe-based sulfide imaging. Consistent with in vitro hypoxia model, SSip-1 and SF-7 AM imaging revealed that supersulfides were decreased and $H_2S$ was increased in the myocardium of the non-scar region of MI model mice (Fig. 4A and B). We next asked whether GSSG improves myocardial dysfunction after MI. To confirm Drp1 S-glutathionylation, the heart was isolated from GSSG-infused mice, and Drp1 S-glutathionylation was detected by immunoprecipitation assay. Drp1 S-glutathionylation was increased in GSSG-infused mouse heart

(Fig. 4C). We previously found that mitochondrial hyperfission but not myocardial senescence occurred in myocardium 1 week after MI, and cilnidipine administration at this time point improves cardiac function and reduces the progression of myocardial senescence[6]. Therefore, GSSG or GSH was continuously administered from 1 week after MI, and cardiac functions were monitored by echocardiography. Cardiac function was partially but significantly improved after GSSG administration (Fig. 4D and E and Supplementary Table 1). Although more than 80% GSH was maintained in reduced form 30 days after preparation (Supplementary Fig. 11A), GSH administration did not improve cardiac function (Fig. 4D and E and Supplementary Table 1). GSSG but not GSH treatment also reduced cardiac hypertrophy 5 weeks after MI (Fig. 4F and Supplementary Table 2). Administration of glutathione ethyl ester

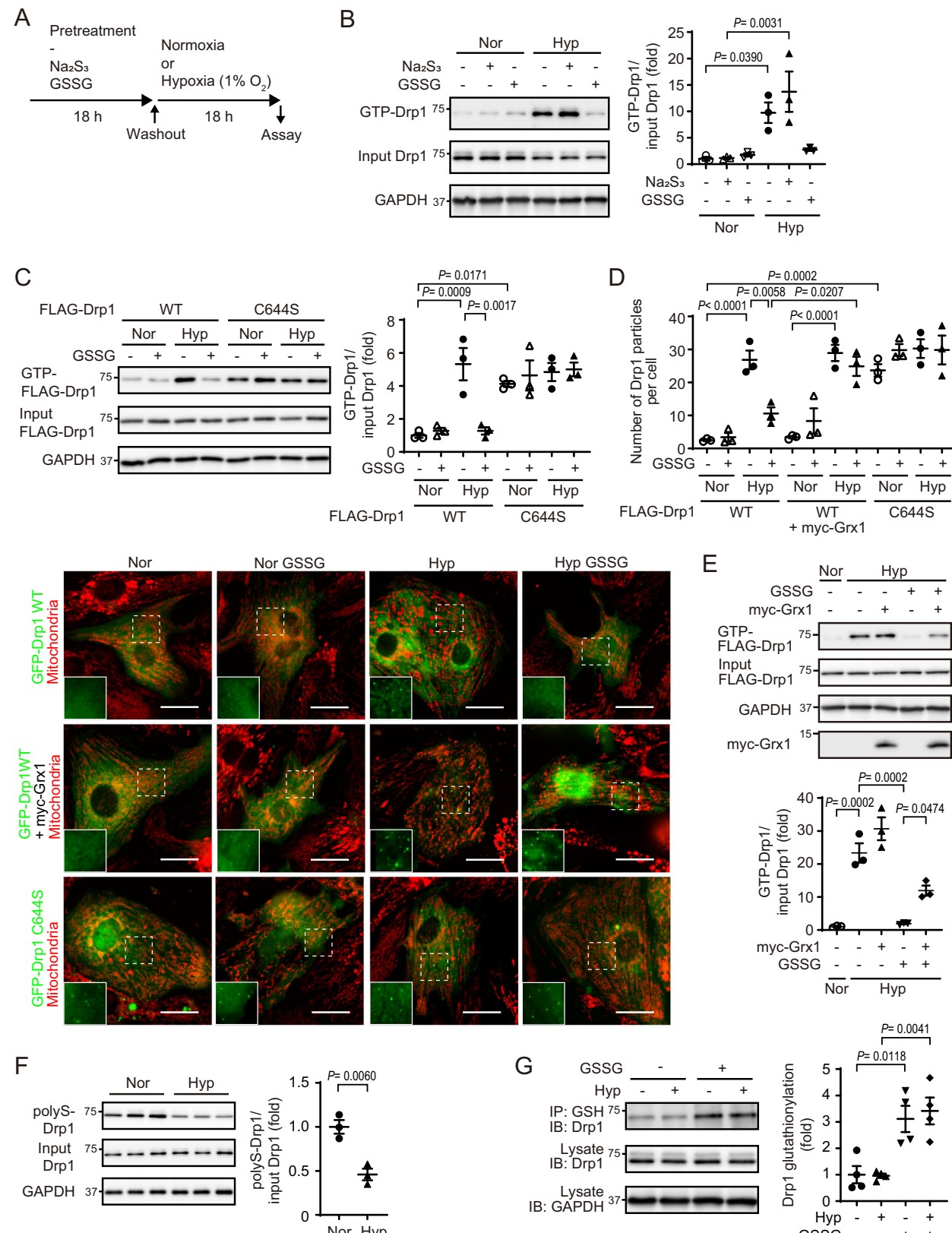

(GEE), membrane-permeable GSH analog, also had no effect on cardiac function and hypertrophy after MI, eliminating concerns about membrane permeability of GSH (Supplementary Fig. 12A-C and Supplementary Table 3 and 4). Administration of GSSG, GSH and GEE did not affect lipid peroxidation and total antioxidant levels in the basal heart (Supplementary Fig. 11B and C). The dose of these types of

glutathione is suitable for heart redox status in the basal state. In addition, GSH but not GSSG administration reduced acute oxidative stress induced by ischemia-reperfusion (Supplementary Fig. 11D and E), suggesting that our condition for GSH administration has adequate antioxidant activity and the antioxidant effect is not sufficient for cardioprotective effects after MI. Next, to analyze how GSSG but not

**Fig. 3 | S-glutathionylated but polysulfidated Drp1 prevents its activation after hypoxic stress. A** Timeline and schedule. After NRCMs were treated with GSSG or Na₂S₃, cells were washed to remove these, and then cultured under normoxia (Nor) or hypoxia (Hyp). **B** Effect of pretreatment of GSSG or Na₂S₃ on the GTP-binding activity of Drp1. ($n = 3$ independent experiments). **C** Effect of GSSG treatment on the GTP-binding activity of Drp1 wild-type (WT) or C644S ($n = 3$ independent experiments). **D** Representative images of GFP-Drp1 (green) and mitochondria (red) in NRCMs treated with or without GSSG under Nor or Hyp. GFP-Drp1 with or without myc-Grx1 was transfected into NRCMs. The dashed square is enlarged and is shown as green single channel. The average number of GFP-Drp1 particles per cell was quantified. ($n = 3$ independent experiments). Scale bars, 20 μm. **E** Effect of Grx1 expression on the inhibition of the GTP-binding activity by GSSG. ($n = 3$ independent experiments). **F** Polysulfidation (PolyS) of Drp1 under Nor or Hyp. ($n = 3$ independent experiments). **G** Effect of hypoxic stress on Drp1 S-glutathionylation. S-glutathionylated Drp1 was immunoprecipitated using an anti-glutathione antibody. ($n = 4$ independent experiments). Data are shown as the means ± SEM. Significance was determined by one-way ANOVA followed by Tukey's post-hoc test or two-sided unpaired t-test **F**. Source data are provided as a Source Data file.

GSH improves cardiac function after MI, we evaluated mitochondrial morphology and function. Electron microscopic analysis showed mitochondrial fragmentation in myocardium from MI-operated mice (Fig. 4G). Complex I activity, the mitochondrial to nuclear DNA (mtDNA / nDNA) ratio and OCR were also decreased in the heart by MI operation (Fig. 4H and I and Supplementary Fig. 13A and B). These mitochondrial dysfunctions were improved by GSSG but not GSH treatment (Fig. 4G – I and Supplementary Fig. 13A). Moreover, impaired complex I activity in MI-operated heart led to high NADH/NAD⁺ ratio referred to as NADH reductive stress (Fig. 4J). This reductive stress was suppressed by GSSG but not GSH treatment. On the other hand, the increase in 4-Hydroxy-2-nonenal (4-HNE), a biomarker of oxidative stress was not inhibited by GSSG and GSH (Fig. 4K). Consistent with mitochondrial functions, the increase in SA-β-gal and p53-positive myocardial senescence in MI-operated heart was also suppressed by GSSG (Fig. 4L and Supplementary Fig. 13C). Cardiac dysfunction is associated with increased risk for multi-organ dysfunction. In our MI model system, effects on liver and kidney functions were limited (Supplementary Fig. 13D-F), but increased blood urea nitrogen (BUN), a kidney function biomarker, in MI-operated mouse was not observed in GSSG-treated groups (Fig. 4M). All these results suggest that oxidized GSSG but not reduced GSH ameliorates supersulfide catabolism-related heart failure by improving mitochondrial functions.

## Drp1 C644S knock-in mouse fails to cardioprotective effect of GSSG

To investigate the pivotal role of polysulfidated Cys644 of Drp1 in ischemic tolerance of the heart and cardioprotective effects of GSSG, we generated Drp1 C644S (Cys to Ser) knock-in mouse using CRISPR Cas9 system (Fig. 5A). In this study, we used heterozygous *Drp1^C644S(CS)/+* mice and wild-type *Drp1^+/+* littermates (Fig. 5B). There are no significant differences in the body weight, water intake, food intake, urine volume and feces volume between *Drp1^+/+* and *Drp1^CS/+* mice (Supplementary Fig. 14A). Heart and kidney weights of *Drp1^CS/+* mice were the same as those of *Drp1^+/+* mice, whereas liver weight of *Drp1^CS/+* mice was higher than that of *Drp1^+/+* mice (Supplementary Fig. 14B). Echocardiographic measurements showed that the basal cardiac functions of *Drp1^CS/+* mice were almost same with those of *Drp1^+/+* mice, excepting smaller LVAWd (left ventricular end-diastolic anterior wall thickness) in *Drp1^CS/+* mice (Supplementary Fig. 14C). Both *Drp1^+/+* and *Drp1^CS/+* mice were subjected to sham or MI surgery. After 1 week, cardiac functions were measured and mice were further subjected to vehicle or GSSG administration groups (Fig. 5C). *Drp1^CS/+* mice showed more severe cardiac dysfunction rather than *Drp1^+/+* mice 1 week after MI (Fig. 5D-F and Supplementary Table 5). Consistent with the highly basal activity of Drp1 C644S mutant (Fig. 3C), *Drp1^CS/+* mice are more vulnerable to ischemic stress. Moreover, GSSG administration improved ejection fraction (EF) and fractional shortening (FS) in MI-operated *Drp1^+/+* mice, whereas cardiac functions of MI-operated *Drp1^CS/+* mice were not recovered by GSSG treatment (Fig. 5G,H and Supplementary Table 6). GSSG treatment also had no inhibitory effect on MI-induced cardiac hypertrophy in *Drp1^CS/+* mice (Fig. 5I). These results suggest the critical role of Drp1 Cys644 in the cardioprotective effects of GSSG.

## Computational relationship between bulkiness of Drp1 Cys644 and conformation

Although AI-based protein structure predictions using AlphaFold2 become powerful tools with high reliability and accuracy, one of the limitations of this approach is the availability of structures and sequences datasets for the prediction. Truly intrinsically disordered regions (IDRs) are modeled as a linker with fragile confidence regions. Thus, the MD simulation can elucidate the molecular mechanism and structure of IDRs. During the MD simulation, the per-residue secondary structure is an influential parameter to analyze the protein structure formation, particularly the IDRs (Supplementary Fig. 15A). The variable domain (VD) in Drp1 that is located between the Middle and the GED domain weakly interacts with the mitochondrial outer membrane-localized cardiolipin[39] and precludes interaction with mitochondrial fission factor (Mff) protein[40,41]. To explore the conformation landscape of the VD in Drp1, the unstructured domain become folded as multiple turns after 50 ns-trajectory according to the secondary structure analysis (Supplementary Fig. 15B). After obtaining a representative model of Drp1 from the MD simulation of CysSH (depolysulfidated Cys644), the model was altered into more 3 forms: CysSSH (polysulfidated Cys644), CW (C644W: polysulfidation-mimic form as previously described[18]), and GSH (S-glutathionylated Cys644) (Supplementary Fig. 15C).

According to the crystal structure (PDB ID: 5WP9)[42], Cys644 resided on helix 7 (H7) revealing the interaction between Glu640 on the same helix and Thr415 located on helix 3 (H3). Structural compactness of the Drp1 was observed through 500 ns-trajectory by the radius of gyration ($R_g$). $R_g$ of alpha Carbon (Cα) of H3, H6, and H7 was evaluated for the 4 systems; CysSH, CysSSH, CW, and GSH based on the all-atom MD simulations (Fig. 6A). The result indicated that CysSH showed the most stable structure over a 500 ns MD simulation by considering the average $R_g$, which was $12.29 \pm 0.20$ Å. reflecting the consolidated compactness of the protein structure. Furthermore, the normalized Root Mean Square Fluctuation (RMSF) analysis calculated from the 100 to 500 ns-MD simulation elucidated the structural stability among the studied systems. The fluctuation pattern of the CysSH was slightly more stable than that of the CysSSH, especially in the VD region, although overall, the two systems were similar. In the cases of the systems with CW and GSH, the H3, VD, and the initial part of the H7 domains were observed to fluctuate. This observation is consistent with the $R_g$ result.

The range of 400–500 ns-trajectories was chosen to analyze the percentage of hydrogen bonding (H-bond) occupancy between the sidechain of Glu640 and bound residue according to the following criteria: the distance of acceptor⋯donor ≤ 3.5 Å and the angle of acceptor⋯H-donor ≥ 120°. We also found that only CysSH showed a moderate %H-bond between Glu640 and Ser637 (Fig. 6B). This interaction might restrict the protein motion and contribute to forming the hidden distribution of Ser637. Because the phosphorylation state of Ser637 is important for Drp1 activity[43-45], differences in Ser637 distribution during 4 systems may reflect Drp1 activity.

To investigate whether Ser637-Glu640 interaction in CysSH is involved in the activity of Drp1, mutation analysis was performed. GTP pull-down assay showed that Drp1 was activated under hypoxia

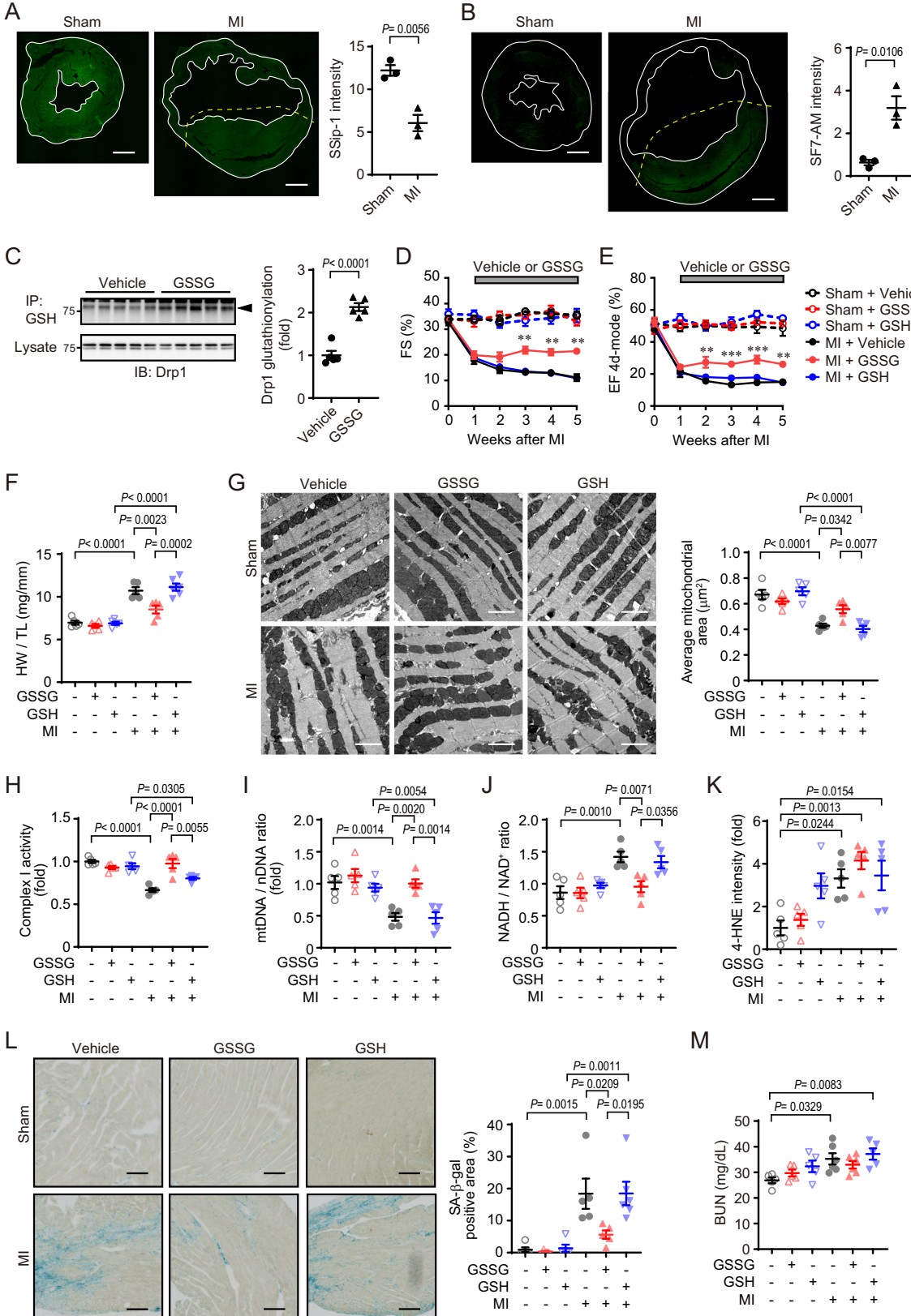

(Fig. 7A). Polysulfidation mimic CW mutant inhibited hypoxia-induced activation, whereas the depolysulfidated C644S (CS) mutant was constitutively activated. E640A (EA) mutant impaired activation under hypoxia. E640A C644S (EA CS) double mutant also lost constitutive activity. Consistent with the activity, the EA mutant inhibited hypoxia-induced interaction with FLNa (Fig. 7B). To better understand this

scenario, MD simulations were additionally performed for EA (E640A with depolysulfidated Cys644) system. Principal component analysis (PCA) was then used to analyze protein dynamics and distribution. Specifically, a total of 4000 snapshots from 100 to 500 ns were evaluated for protein motion in the VD region, which is responsible for binding to FLNa. The results were compared to those of the CysSH and

**Fig. 4 | GSSG but not GSH administration improves cardiac function after MI through Drp1 S-glutathionylation. A, B** Representative images of supersulfide **A** and hydrogen sulfide **B** in mouse heart slice 5 weeks after MI. (*n* = 3 mice per treatment). Supersulfides and hydrogen sulfide were detected using SSip-1 and SF7-AM probes, respectively. The yellow dashed line shows the border of the scar and non-scar area. Scale bars, 1 mm. **C** Drp1 S-glutathionylation in mouse heart at 3 days after GSSG administration. Bands corresponding to Drp1 S-glutathionylation (arrowhead) were quantified (*n* = 5 mice per treatment). **D, E** Changes in fractional shortening (FS) **D** and ejection fraction (EF) **E** in mice after MI. An osmotic pump filled with saline, GSSG or GSH (30 mg/kg/day) was implanted intraperitoneally at 7 days after MI. (*n* = 6 mice for MI + GSH, *n* = 5 for others). **F** Effect of GSSG or GSH on heart weight (HW) / tibia length (TL) ratio in mice 5 weeks after MI. (n = 6 mice

for MI + GSH, *n* = 5 for others). **G** Representative electron micrographs of peri-infarct zone myocardium. The average size of mitochondrial area was quantified. (*n* = 5 mice per treatment). Scale bars, 2 μm. **H–K** Quantification of complex I activity of mitochondrial supercomplex (**H**), mtDNA/nDNA ratio (**I**), NADH/NAD⁺ ratio **J**, 4-HNE intensity **K** in myocardium. (*n* = 5 mice per treatment). **L** Effect of GSSG or GSH on the proportion of SA-β-gal-positive area in peri-infarct zone myocardium 5 weeks after MI. (*n* = 6 mice for MI + GSH, *n* = 5 for others). Scale bars, 200 μm. **M** Plasma BUN levels measured 5 weeks after MI. (*n* = 5 mice per treatment). Data are shown as the means ± SEM. Significance was determined by two-sided unpaired t-test **A–C**; one-way ANOVA followed by Tukey's post-hoc test **F–M**; two-way ANOVA followed by Sidak's post-hoc test **D, E**. Source data are provided as a Source Data file.

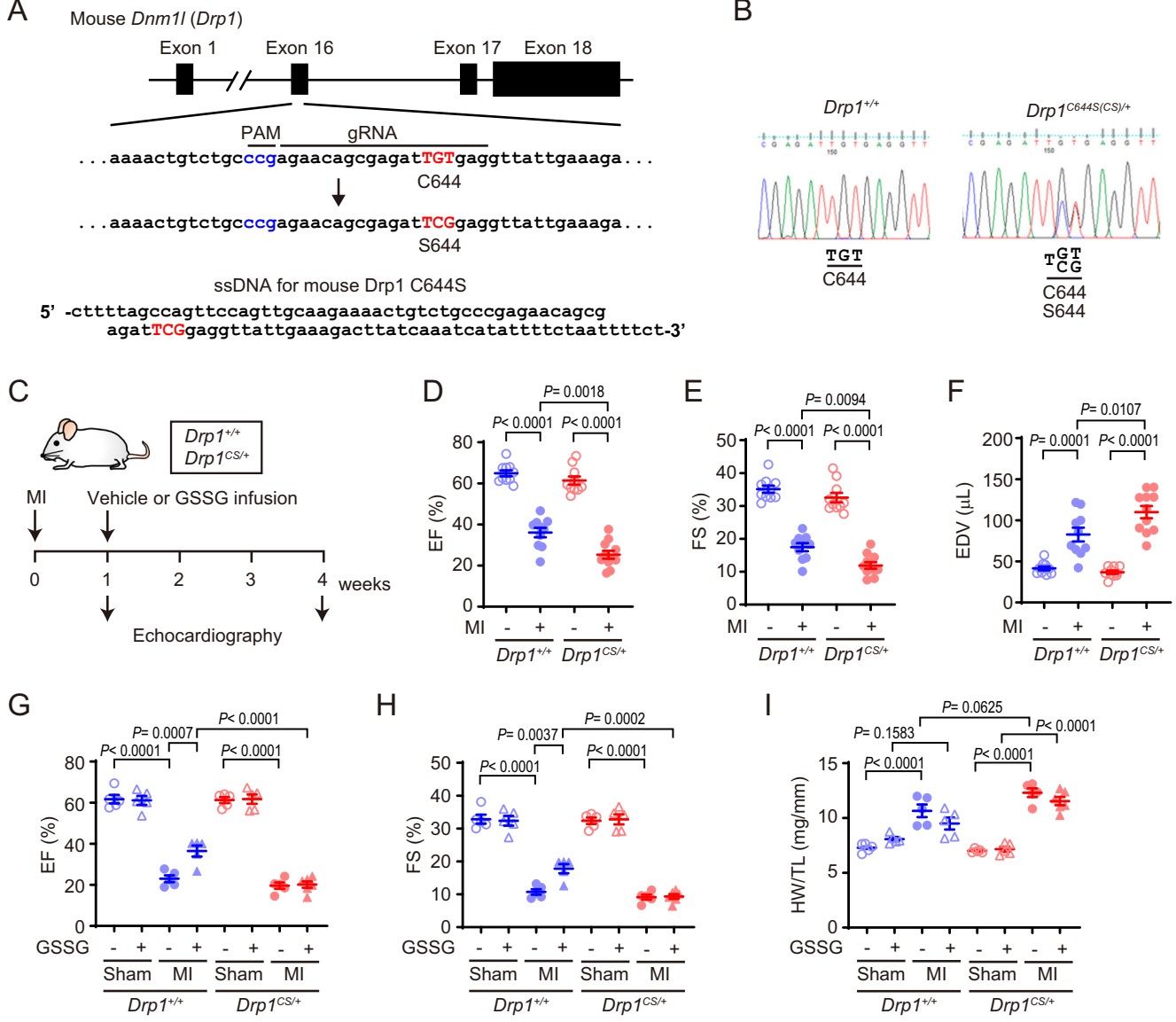

**Fig. 5 | Drp1 C644S hetero knock-in mice fail to cardioprotective effects of GSSG after MI. A** Schematic illustration of CRISPR/Cas9-mediated gene editing for Dnm1l (Drp1). Drp1 C644S knock-in mice were generated by mutating Cys TGT codon at position 644 to Ser TCG codon. The sequences of gRNA and template DNA were shown. **B** Genome DNA sequencing identified *Drp1^C644S(CS)/+* hetero knock-in mutation. **C** Experimental protocol. One week after myocardial infarction (MI), cardiac functions of *Drp1^+/+* and *Drp1^CS/+* were analyzed by echocardiography. Then, an osmotic pump filled with saline or GSSG (30 mg/kg/day) was implanted

intraperitoneally, and cardiac functions were analyzed again after an additional 3 weeks. **D–F** Changes in ejection fraction (EF) **D**, fractional shortening (FS) **E** and end-diastolic volume (EDV) **F** in *Drp1^+/+* or *Drp1^CS/+* mice 1 week after MI. (*n* = 10 mice for *Drp1^+/+* MI, *n* = 11 mice for *Drp1^CS/+*) **G–I** Effect of GSSG on EF **G**, FS **H** and heart weight (HW) / tibia length (TL) ratio **I** in *Drp1^+/+* or *Drp1^CS/+* mice 4 weeks after MI. (*n* = 6 mice for *Drp1^CS/+* MI + GSSG, *n* = 5 for others). Data are shown as the means ± SEM. Significance was determined by one-way ANOVA followed by Tukey's post-hoc test. Source data are provided as a Source Data file.

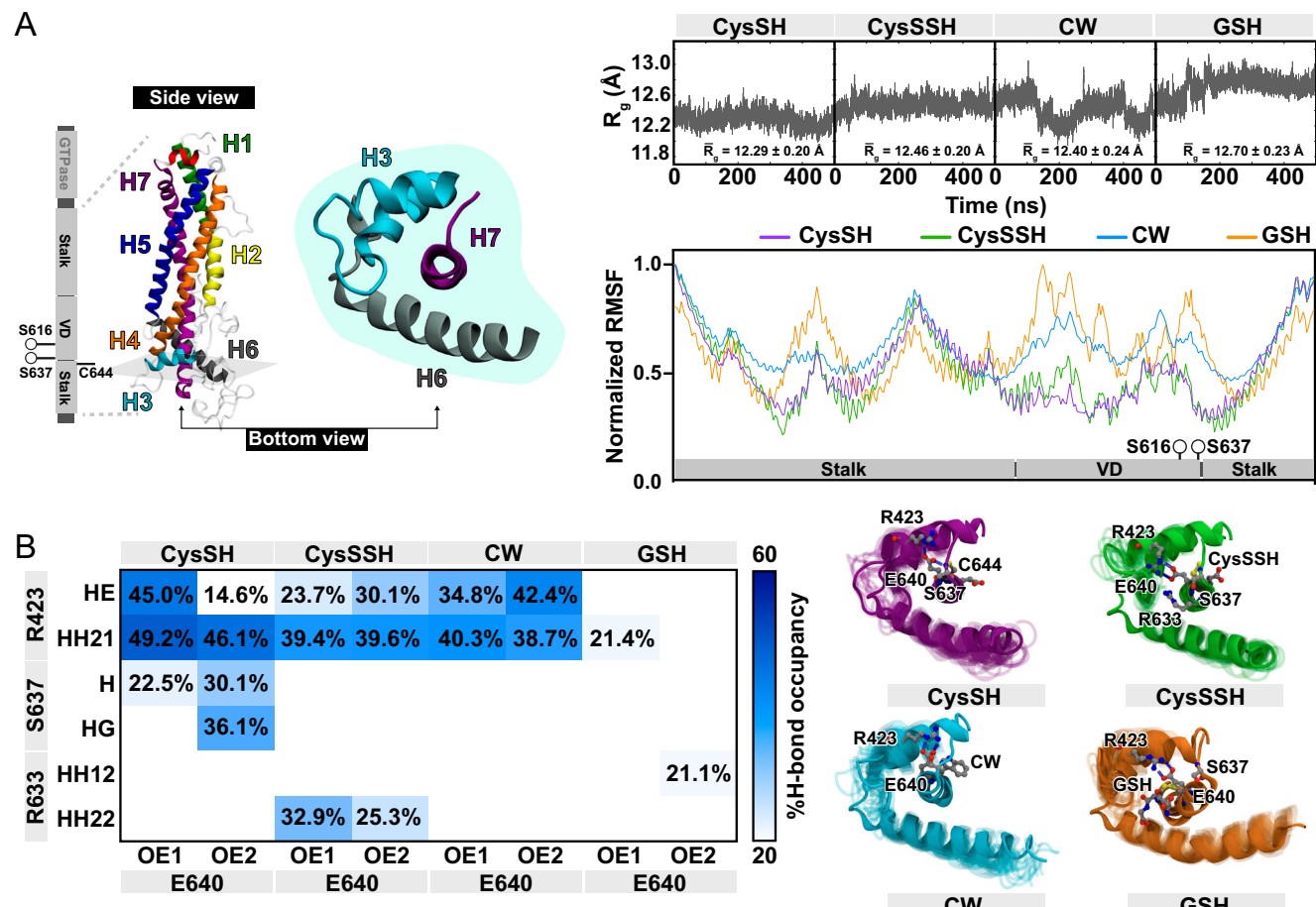

**Fig. 6 | Computational relationship between bulkiness of Drp1 at Cys644 and structural conformation. A** The comparison of $R_g$ of Cα-H3, Cα-H6, and Cα-H7 of Drp1 was analyzed from 0 to 500 ns-trajectories and structural stability was analyzed from 100 to 500 ns-trajectories. CysSH: depolysulfidated Cys644, CysSSH: polysulfidated Cys644, CW: Cys644Trp polysulfidation mimic form, GSH: S-glutathionylated Cys644. **B** Hydrogen bonding interaction was depicted in terms of %occupancy shown as a grid plot ranging from 20 (white) to 60 (blue). Only %H-bond occupancy with ≥ 20 are labeled.

CysSSH systems. The first two PCs (PC1-PC2) 3D positional coordinates of each system were plotted on a scatter graph to visualize the dynamical protein distribution (Fig. 7C). The protein distribution of the CysSH and EA systems showed a similar pattern at the VD region. However, the EA system exhibited two distinct clusters. The CysSSH system had a narrower distribution than other systems, suggesting that not only the Glu640 residue but also the influence of polysulfidation at Cys644 may contribute to the regulation of the VD domain. The E640A mutation may result in the loss of interaction between Glu644 and Ser637, which can lead to an inhibitory effect on hypoxia-induced activation. These results suggest that the Ser637-Glu640 interaction that is predicted from the MD simulation is critical for Drp1 activation and Cys644 bulkiness by S-glutathionylation and polysulfidation reduces Drp1 activity by disrupting the Ser637-Glu640 interaction.

## Discussion

Because the heart needs a lot of energy for continuous beating, more than 40% of the cytosolic space of cardiomyocytes is occupied by mitochondria. Therefore, disruption of mitochondrial quality has a critical role in the progression of heart disease. We previously reported that Drp1-mediated mitochondrial fission triggers myocardial senescence in MI model mice and Drp1 forms a ternary complex with FLNa and actin for its activation in the mitochondrial fission site[6]. In this study, we found that supersulfide catabolism is a major cause of mitochondrial and cardiac dysfunction. Supersulfide imaging assay

showed that hypoxic and pseudohypoxic stress induce supersulfide catabolism (Fig. 4 and Supplementary Fig. 9 and 10). Additionally, our previous work shows that Echinochrome A improves cardiac dysfunction after MI with the suppression of supersulfide catabolism[15], indicating the correlation between supersulfide catabolism and heart failure progression. Polysulfidation assay showed that hypoxic and pseudohypoxic stress leads to supersulfide catabolism-mediated Drp1 depolysulfidation (Fig. 3 and Supplementary Fig. 10). Drp1 depolysulfidation has been also observed in the pressure overload and the isoprenaline-induced heart failure model[18,30]. Depolysulfidated Drp1 induces mitochondrial fission by promoting the complex formation with FLNa[18]. $Drp1^{CS/+}$ knock-in mice experiments suggested that Cys644 polysulfidation is important for ischemic tolerance of the heart (Fig. 5). GSSG preferentially S-glutathionylated Drp1 protein at Cys644 polysulfides rather than other Cys thiols (Fig. 1). A negatively charged-deprotonated thiol group of Cys can react with electrophiles including GSSG[46]. Because the pKa value of CysS_(n)SH ($n ≥ 1$) is computationally speculated to be 4.3 whereas that of CysSH is 8.3[47], the polysulfidated thiol group of Cys644 is easily deprotonated and preferentially reacted with GSSG for S-glutathionylation (Fig. 1 and Supplementary Fig. 3). S-glutathionylated Drp1 at Cys644 prevented Drp1-FLNa complex formation and supersulfide catabolism-mediated cardiac dysfunction in vivo mice model (Fig. 7D).

In this study, we showed that GSSG but not GSH has a cardioprotective effect after MI (Fig. 4). Molecular mechanism explaining the differences between GSSG and GSH treatment is a major focus of this

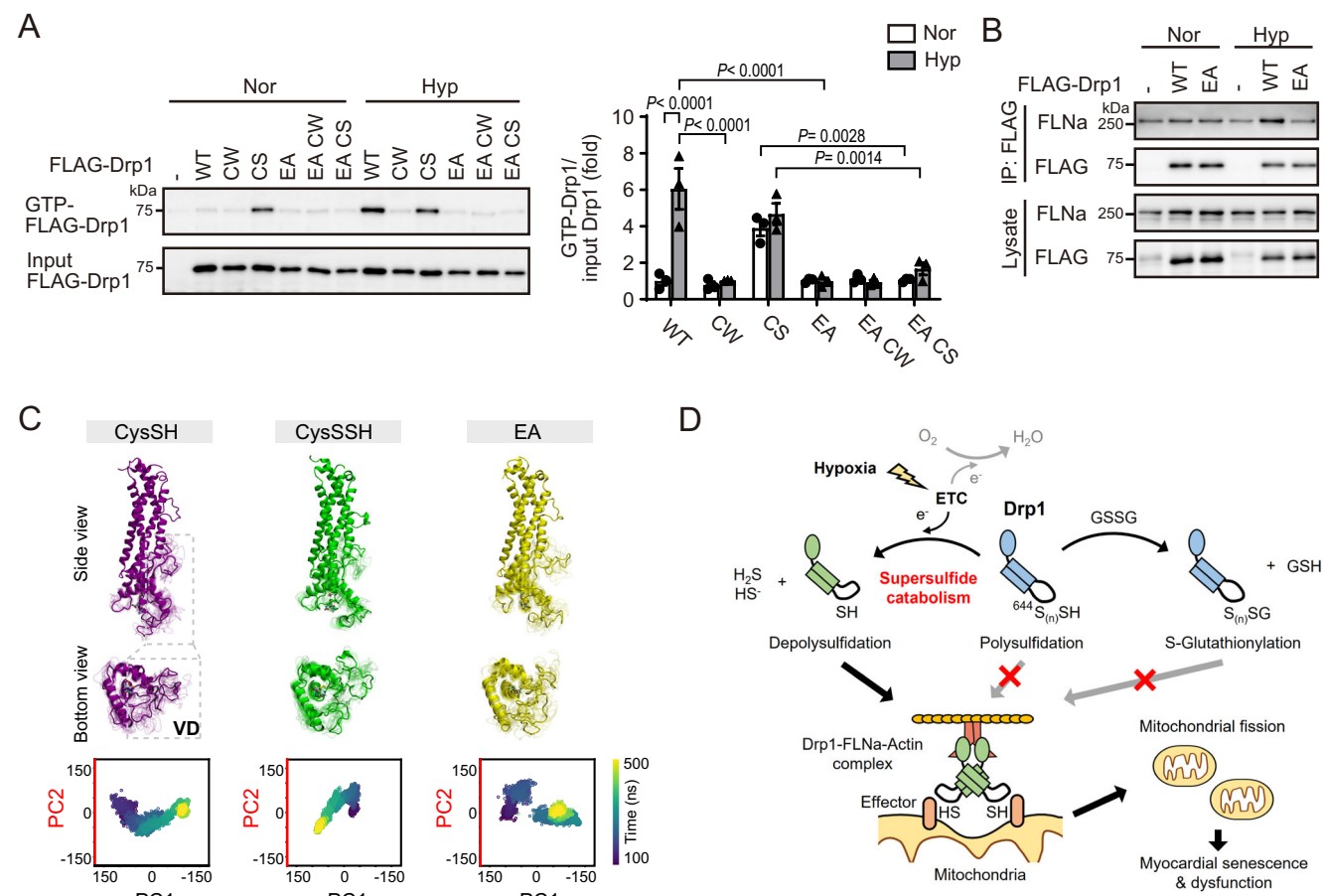

**Fig. 7 | Cys644 bulkiness mediated Glu640-Ser637 interaction regulates hypoxia-induced Drp1 activation. A** The GTP-binding activity of exogenous FLAG-Drp1 mutant under normoxia (Nor) or hypoxia (Hyp) ($n$ = 3 independent experiments). WT: wild-type, CW: C644W, CS: C644S, EA: E640A, EA CW: E640A C644W, EA CS: E640A C644S. **B** Effect of E640A mutation on hypoxia-mediated interaction between FLNa and FLAG-Drp1. FLAG-Drp1 was immunoprecipitated from 293 T cells transfected with FLAG-Drp1 WT or EA ($n$ = 3 independent experiments). **C** Structural dynamic comparison of variable domain (VD) among CysSH, CysSSH and EA was analyzed using principal component analysis (PCA). **D** Schematic of supersulfide catabolism-mediated mitochondrial fission and its protective role by GSSG. Electrons from mitochondrial ETC under hypoxia mediate catabolism of the polysulfide group at Drp1 Cys644 (Cys644-$S_{(n)}$SH, $n \geq 1$). Depolysulfidated Drp1 induces mitochondrial fission through Drp1-FLNa-Actin complex formation, resulting in myocardial dysfunction. GSSG mediates S-glutathionylation of Drp1 at Cys644, preventing supersulfide catabolism-induced mitochondrial fission. Data are shown as the means ± SEM. Significance was determined by one-way ANOVA followed by Tukey's post-hoc test **A**. Source data are provided as a Source Data file.

study. GSSG but not GSH administration improved mitochondrial hyperfission and bioenergetics after MI (Fig. 4G and H and Supplementary Fig. 13A), suggesting that the protection of mitochondrial quality is the primary target of GSSG for cardioprotection. To analyze the mechanistic connection between Drp1 Cys644 S-glutathionylation and cardioprotection, we generated Drp1 C644S knock-in mouse, and elucidated that Drp1 Cys644 is critical for ischemic tolerance of cardiomyocytes and cardioprotective role of GSSG in vivo (Fig. 5). Identification of Drp1 Cys644 S-glutathionylation from heart would provide a more compelling mechanistic link between Cys644 S-glutathionylation and cardioprotection. We attempted to purify endogenous Drp1 from hearts for mass spectrometry analysis, but no antibodies were available to sufficiently immunoprecipitate endogenous Drp1 as a limitation of this study. Therefore, we quantitatively analyzed polysulfidation and S-glutathionylation of Drp1 using mass spectrometry analysis of exogenously expressed Drp1. Nearly 40% of Cys644 of Drp1-FLAG from HeLa cells was polysulfidated (Fig. 1G). Reconstitution assay using depolysulfidated Drp1 from *E. coli* showed that Na₂S₄ treatment highly increased polysulfidation at Cys644 compared with other cysteines, and this polysulfidated Cys644 was preferentially S-glutathionylated by GSSG (Fig. 1J and Supplementary

Fig. 3). Additionally, cell-based assay also supports the critical role of Cys644 S-glutathionylation on GSSG-mediated mitochondrial protection (Fig. 3). These pieces of evidence suggest that GSSG protects cardiomyocytes from hypoxic stress by maintaining mitochondrial quality through Drp1 S-glutathionylation rather than GSH-mediated antioxidant effect.

It has been reported that GSSG administration slightly increased the GSH levels in plasma and some organs such as liver and lung[48], and decreased lipid peroxidation production in liver[49]. This previous evidence would speculate that either GSH and GSSG administration protects organs through antioxidant effects. However, antioxidant effects of GSSG have been reported when used at high concentrations. We used more than 10 times lower dose (30 mg/kg/day) of GSSG and GSH in this study compared with the previous report[49], and found that GSH but not GSSG administration has an ability to suppress acute oxidative stress (Supplementary Fig. 11D,E). This result suggests that the antioxidant effect of GSH is not sufficient for cardioprotective effects, and GSSG protects the heart with mitochondrial protective effects rather than antioxidant effects. However, glutathione (GSH/GSSG balance) system plays a central role in regulating redox homeostasis, and its imbalance is closely related to various types of diseases.

Although GSSG treatment did not affect lipid peroxidation and anti-oxidant activity in basal heart (Supplementary Fig. 11B,C), it will be important to evaluate the effect of GSSG administration on redox balance including ROS and sulfur metabolites in whole body in detail as we assess clinical potentials of GSSG in the future.

Proper redox balance is critical for the maintenance of physiological homeostasis and its abnormality is closely linked to the progression of various diseases[7]. Oxidative stress which is defined as the excess production of ROS and reactive nitrogen species (RNS) relative to antioxidant defense systems can cause myocardial remodeling and inflammation, contributing to heart failure progression. The heart develops multiple antioxidant systems including catalase, thioredoxin/thioredoxin-reductase, superoxide dismutase, and glutathione per-oxidase for preserving redox equilibrium[11]. Excessive antioxidant generation leads to reductive stress, also contributing to pathological myocardial remodeling and heart failure[50]. Especially, reduced form of nicotinamide adenine dinucleotide (NADH) is known as a key molecule of reductive stress and is closely associated with multiple chronic disorders[51,52]. Proper $NAD^+$ / NADH balance is critical for cellular metabolism and mitochondrial energy production. Excessive NADH triggers multiple alterations in energy metabolism and gene transcription as a NADH reductive stress[53]. Additionally, expression of glucose 6 phosphate dehydrogenase and a point mutant of αB-crystallin elevate NADH and GSH levels, which are closely related to reductive stress-induced cardiomyopathy[54–56]. Actually, 5 weeks after MI, $NADH/NAD^+$ ratio was increased in the heart, indicating reductive stress condition (Fig. 4J). In this paper, we showed that accelerated supersulfide reduction to $H_2S$ under hypoxia and pseudohypoxia causes myocardial dysfunction through Drp1 depolysulfidation (Fig. 2 and Supplementary Fig. 10). Although the detailed mechanism of supersulfide reduction to $H_2S$ under hypoxia and pseudohypoxia is still unclear, mitochondrial membrane depolarization would accelerate supersulfide catabolism (Supplementary Fig. 9). Supersulfides such as CysSSH can receive electrons from mitochondrial ETC and is converted to $HS^-/H_2S$ for proper mitochondrial bioenergetics[13]. The disruption of mitochondrial membrane potential may alter the balance of electron acceptance in ETC by oxygen and supersulfides and abnormally enhance supersulfide catabolism. Converted $H_2S$ is oxidized back to persulfides by sulfide:quinone oxidoreductase (SQOR)[57], and accumulation of $H_2S$ by SQOR mutant increases hypoxic brain injury[36]. These pieces of evidence suggest that the balance of super-sulfide metabolism has pivotal roles in redox homeostasis, and excess supersulfide reduction (catabolism) as reductive stress causes mito-chondrial abnormality and cardiac dysfunction.

Using mass spectrometry and biochemical analysis, we identified that Drp1 Cys644 is highly polysulfidated in the basal state, and GSSG treatment increased S-glutathionylation of Cys644 up to approximately 40% (Fig. 1). Cys644 is a redox-active cysteine, and various redox modifications including S-nitrosylation and oxidation have been reported[27–29]. Although we were unable to detect S-nitrosylation in our detection system (Fig. 1G), we do not intend to deny previous reports about the importance of S-nitrosylation and oxidation of Cys644. Previous reports have been able to assess S-nitrosylation and oxidation of endogenous Drp1, albeit indirectly by using a biotin switch assay[27,28]. In contrast, we used Drp1 overexpressed in HeLa cells for mass spec-trometry. It is quite possible that the modification state differs between endogenous and exogenous Drp1. Additionally, S-nitrosylation and oxidation have been identified from the brain[27,28]. Organ and cell-type may be important for variation of modification status of Drp1 Cys644.

Some preclinical mouse model experiments have reported car-dioprotective effects of GSH administration[58,59], whereas GSH failed to improve cardiac contractility in our mouse MI model (Fig. 4). This discrepancy may reflect the differences in experimental models. The successful effect of GSH administration has been mainly evaluated in acute and strong oxidative stress models such as ischemia-reperfusion[58–60]. On the other hand, we tested the effect of GSH administration on the chronic MI model. GSSG and GSH were administrated 7 days after MI which acute oxidative stress would calm. Actually, cardioprotective effects of GSH administration depend on the duration of ischemia, and more than 40 min hypoxia decreases cardiac recovery by exogenous GSH in ex vivo hypoxia-reoxygenation heart model[58]. GSH administration might be more effective against the acute oxidative stress model. GSSG improves supersulfide catabolism-related mitochondrial dysfunction through Drp1 S-glutathionylation, protecting cardiomyocytes, and released GSH would also protect the heart from oxidative stress. Therefore, GSSG might preferentially improve chronic heart failure rather than GSH. Clinical trials of anti-oxidant therapy are almost failed[8–10]. Future studies focusing on supersulfide catabolism and S-glutathionylation by GSSG may con-tribute to our understanding of a therapeutic strategy for heart failure and other mitochondria-related diseases.

## Methods

### Reagents and antibodies
Oxidized glutathione, L-reduced glutathione, and proximity ligation assay kit were purchased from Sigma. Sodium trisulfide ($Na_2S_3$) was from Dojindo. SSip-1 DA was from Goryo Chemical. SF7-AM was from Cayman. Collagenase II was from Worthington. Senescence β-Galactosidase Staining Kit was from Cell Signaling. Anti-Drp1 (H-300) and anti-FLNa (E-3) antibodies were from Santa Cruz. Anti-sarcomeric alpha-actinin (EA-53) and Anti-s-glutathione (D8) anti-bodies were from Abcam. Anti-DLP1 (Drp1) antibody was from BD biosciences. Alexa-conjugated anti-mouse and rabbit IgG were from Thermo Fisher.

### Animals
All protocols using mice and rats were reviewed and approved by the ethics committees at National Institute for Physiological Sciences or the Animal Care and Use Committee, Kyushu University, and were performed according to the institutional guidelines concerning the care and handling of experimental animals. Seven-week-old male C57BL6J mice and Sprague-Dawley (SD) rats were purchased from Japan SLC, Inc. (Shizuoka, Japan). All mice were kept in plastic cages in a climate-controlled animal room with a 12-hour light/dark cycle, and then male mice aged 8-10 weeks were used for experiments.

### MI surgery and transthoracic echocardiography
MI surgery was performed on 8 to 10-week-old male C57BL/6 J mice. MI was artificially induced by permanent ligation of the left anterior descending (LAD) coronary artery. The LAD artery was ligated 2 to 3 mm distal to the left atrial appendage using an 8-0 silk suture. The intercostal space, pectoralis major muscle, and skin were closed sequentially using a 5-0 silk suture. Acute myocardial ischemia was deemed successfully induced when blanching of the anterior left ventricle distal to the ligature and a significant ST-segment elevation, confirmed by later echocardiography, were observed. Endotracheal intubation was performed prior to thoracotomy to facilitate mechan-ical ventilation of the mice. For the sham group, mice underwent the same procedure except for LAD ligation[6]. All surgical procedures were conducted on mice anesthetized with 3.0% isoflurane (Pfizer) mixed with air for 30 sec, followed by maintenance at 2.0% for MI surgery or 1.0% for echocardiography. GSH was solved in saline with 1 mM EDTA for the stability of thiol groups[61]. A mini-osmotic pump (Alzet) filled with vehicle (saline), GSSG, GSH or GEE (30 mg/kg/day) was implanted intraperitoneally into mice at 7 days after MI. Transthoracic echo-cardiography was performed using the Vevo3100 imaging system (FUJIFILM VisualSonics) before or every week after the operation. Left ventricle fractional shortening (FS) was acquired from a parasternal short-axis view of motion-mode echocardiography. 4D-mode left

ventricle ejection fraction (EF 4d-mode) was calculated according to 3D geometry with dynamic motion of the left ventricle myocardium.

## Tissue analysis

At 5 weeks after MI, mouse hearts were removed, washed in PBS, and fixed with 4% PFA in PBS for 12 h at 4 °C. Tissues were cryopreserved with 10, 20 and 30% sucrose in PBS every 12 h at 4 °C, and then embedded in optimal cutting temperature (O.C.T.) compound (Sakura Finetek) and snap-frozen in a bath of cooled 2-methylbutane with crushed dry ice. Twelve-μm-thick sections were cut on a cryostat and used for senescence-associated β-gal staining and supersulfides and hydrogen sulfide staining. Senescence-associated β-gal staining was performed using the Senescence β-Galactosidase Staining Kit (Cell Signaling Technology) according to the manufacturer's instructions with some modifications. β-gal staining solution was used at a final pH of 5.0. SA-β-gal positive area in the peri-infarct zone was quantified. For supersulfides[37] and hydrogen sulfide[35] staining, the heart slice was stained with 10 μM SSip-1 (supersulfides) in HBSS including 0.1% BSA and 0.02% Cremophor EL or 5 μM SF7-AM (hydrogen sulfide) in HBSS for 45 min at room temperature. The sample was washed three times with PBS and then mounted with ProLong Diamond Antifade Mountant (Thermo). The sample was observed using a BZ-X710 microscope (Keyence). The fluorescent intensity in the non-infarct zone (including the peri-infarct zone) was quantified. MDA and total antioxidant capacity of heart homogenate were measured by MDA assay kit (Dojindo) and OxiSelect total antioxidant capacity assay kit (CELL BIOLABS) according to the manufacturer's instructions, respectively.

## Transmission electron microscopy (TEM)

Mouse left ventricular (for sham) and peri-infarct zone (for MI) were cut and prefixed with 2% paraformaldehyde solution containing 0.15 M sodium cacodylate and 2 mM CaCl₂ (pH 7.4) for 3 h on ice and re-cut into 1- to 2-mm cubes. After washing with 0.15 M cacodylate solution, the tissue blocks were immersed in a solution containing 2% osmium tetroxide, 1.5% potassium ferrocyanide, 0.15 M sodium cacodylate, and 2 mM CaCl₂ (pH 7.4) for 1.5 h at room temperature. After washing with distilled water, the tissue blocks were immersed in thiocarbohydrazide (0.01 mg/ml) solution for 40 min and post-fixed with 2% osmium for 1 h. En bloc staining was performed by sequentially immersing the tissue blocks in a solution of 1% uranium acetate overnight at 4 °C and an aqueous solution of lead aspartic acid for 60 min with oven-drying. After dehydration in a graded ethanol series and acetone, the specimens were embedded in durcupan resin. The surface (thickness of 70 nm) of the resin-embedded tissue was exposed using a diamond knife on an Ultracut UC7 (Leica Microsystems). TEM samples were imaged with a Veleta CCD camera (Olympus) equipped on a JEOL1010 microscope (JEOL). The mitochondrial area was measured and quantified from at least 100 mitochondria per sample using ImageJ.

## Voluntary exercise

The physiological spontaneous activity of mice was measured using an animal movement analyzing system (ACTIMO-100, Shinfactory). For voluntary exercise, 9-week-old male C57BL/6 J mice were bred for 4 months in a rectangular enclosure cage (30 ×20 cm) with a running wheel equipped to the wall. The spontaneous walking activity of exercise mice is about 10 km/day, which is 10 times higher than sedentary mice[62].

## Mitochondrial analysis from tissue sample

To analyze mitochondrial complex I activity, mitochondria was purified from frozen mouse heart tissue[63]. Isolated heart tissue was homogenized in ice-cold Isolation Buffer (10 mM Tris pH7.4, 200 mM Sucrose and 1 mM EGTA) using Dounce homogenizer. Samples were centrifuge at 600 x g for 10 min at 4 °C to precipitate cell debris. Supernatant was centrifuged at 7000 x g for 10 min at 4 °C, and the

pellet was resuspended in ice-cold Isolation Buffer. Protein concentration of samples was measured, and 50 μg samples were centrifuged again at 7000 x g for 10 min at 4 °C. Pellet was suspended in 20 μl of NativePAGE Sample Buffer (Thermo Fisher) containing 2% digitonin. Samples were subjected to blue native-PAGE (BN-PAGE) using NativePAGE Running Buffer Kit (Thermo Fisher). BN-PAGE gel was incubated in the Complex I substrate solution (2 mM Tris pH7.4, 0.1 mg/ml NADH and 2.5 mg/ml Nitrotetrazolium Blue chloride) for 20 min and then the reaction was stopped with 10% acetic acid. Gel was scanned and violet bands were quantified as the complex I activity. For mtDNA copy number analysis, total DNA was prepared from frozen mouse heart tissue using QIAamp DNA Mini Kit (Qiagen). The ratio of mtDNA/nDNA was measured by quantitative PCR. Primers for mouse mtDNA (F: CTAGAAACCCCGAAACCAAA, R: CCAGCTATCACC AAGCTCGT) and mouse nDNA (hexokinase 2) (F: GCCAGCCTCTCCT GATTTTAGTGT, R: GGGAACACAAAAGACCTCTTCTGG) were used.

## GSSG/GSH ratio and NADH/NAD⁺ ratio analysis

GSSG and GSH levels were quantified using GSSG/GSH Quantification Kit (Dojindo) according to the manufacturer's instructions. The heart tissues from sedentary and exercised mice were snap-frozen in liquid nitrogen. The tissues (20 mg) were homogenized in 100 μl of 5% 5-Sulfosalicylic acid using BioMasher II (Nippi). NADH/NAD⁺ ratio was quantified using NAD/NADH Assay Kit-WST (Dojindo) according to the manufacturer's instructions.

## Analysis of liver and kidney functions

For serum biochemical analysis, serum levels of BUN, AST and ALT were measured using Fuji Dry-Chem NX5000 (Fujifilm Medical). Neutrophil gelatinase-associated lipocalin (NGAL) expression levels in kidney were measured using Mouse NGAL ELISA kit (Proteintech) according to the manufacturer's instructions. Kidney tissues were homogenized in PBS (10 μl per mg tissue) and freeze-thawed three times. Lysates were centrifuged (16,000 x g for 10 min at 4 °C) and 10,000 diluted supernatant was used for ELISA assay. For kidney fibrosis, paraffin-embedded kidney samples were stained with Picro-Sirius Red and fibrosis area was quantified.

## Cell culture and transfection

NRCMs were prepared from the ventricles of 2-day-old SD rats as follow. Rat pups were sacrificed and the left ventricles were removed and minced. The minced tissue was pre-digested in 0.05% trypsin-EDTA (Gibco) 18 h at 4 °C and then digested in 1 mg/ml collagenase type 2 (Worthington) in PBS for 25 min at 37 °C. The dissociated cells were plated in a 10-cm culture dish and incubated at 37 °C in a humidified atmosphere (5% CO₂, 95% air) for 1.5 h in DMEM (low glucose) containing 10% FBS and 1% penicillin and streptomycin. Floating cells were collected and plated into matrigel-coated culture dishes or glass bottom dishes at a density of around $2–3 × 10^5$ cells/ml. After 24 h, the culture medium was changed to DMEM containing 2% FBS. Cells were starved for 1 day in DMEM with 5 mM Taurine for experiments. Human iPS cell-derived cardiomyocyte product, iCell Cardiomyocytes² was purchased from Cellular Dynamics and maintained according to the manufacturer's instructions. HeLa (ATCC Cat#CCL-2) and 293 T (ATCC Cat#CRL-3216) cells were cultured in DMEM (high glucose) supplemented with 10% FBS. For hypoxia experiment, cells were exposed to hypoxia (1% O₂) in 5% CO₂ humidified multigas incubator. O₂ concentration in the incubator is controlled by injection of nitrogen gas. Plasmid DNA was transfected using Viafect (Promega) for HeLa and 293 T cells.

## Measurement of protein polysulfidation by IAA-biotin assay

To detect polysulfidated proteins, we used a modified biotin switch assay using iodoacetyl-biotin (IAA-biotin)[64,65] that was originally reported by Doka et al.[66]. Cell lysates (1.5 mg/ml, 150 μl) were prepared

using IAA assay buffer (40 mM phosphate pH7.0, 150 mM NaCl, 1% Triton X-100, 0.1% SDS, and 3 mM tyrosine) with protease inhibitors (10 μg/ml pepstatin A and 2 μg/ml aprotinin) and 100 μM Iodoacetyl-PEG2-Biotin (Thermo) and incubated at 37 °C for 30 min, and then free probes were removed by desalting columns. The biotinylated proteins were pulled down using NeutrAvidin-agarose beads (Thermo). The beads were washed three times with IAA assay buffer and twice with wash buffer (40 mM phosphate pH7.0, 150 mM NaCl, 1% Triton X-100, and 0.1% SDS) and polysulfidated proteins were eluted by elution buffer (wash buffer with 40 mM DTT). Eluted proteins were subjected to SDS-PAGE.

## Western blot
Samples were subjected to SDS-PAGE and transferred onto PVDF membranes. Membranes were blocked in 2% BSA in Tris-buffered saline with Tween-20 (TBS-T), incubated with primary antibody (Supplementary Table 7 for antibody information and dilution) overnight at 4 °C, followed by corresponding HRP-conjugated secondary antibody. Blots were developed with Clarity Max Western ECL Substrate (Bio-rad) and detected using ImageQuant LAS4000 (Cytiva). Protein band intensity was measured by ImageQuant TL software. Uncropped blot data is shown as Supplementary Fig. 16–20.

## Immunoprecipitation
Protein S-glutathionylation was evaluated by immunoprecipitation assay using anti-s-glutathione antibody[23,67]. Mouse hearts were homogenized using Physcotron (Microtec) in ice-cold lysis buffer (20 mM Hepes pH7.4, 150 mM NaCl, 1 mM EDTA, 3 mM tyrosine, 1% Triton X-100, 0.1% SDS and protease inhibitor cocktail). Tyrosine was added to protect S-S bond of S-glutathionylation from oxidants[68]. The lysate was ultra-centrifuged (100,000 x g for 30 min at 4 °C), and the supernatant was recovered. The protein concentration of the lysate was measured and adjusted to 2 mg/ml with wash buffer (20 mM Hepes pH 7.4, 150 mM NaCl, 1 mM EDTA, 3 mM tyrosine and 1% Triton X-100). The aliquot of the lysate (300 μg) was incubated with 1 μl of anti-s-glutathione antibody and 10 μl of Protein G Sepharose™ 4 Fast Flow (Cytiva) overnight at 4 °C. The beads were centrifuged (1000 x g for 1 min at 4 °C) and washed three times with wash buffer. The bound proteins were eluted by 2x Laemmli buffer without reducing reagent and subjected to SDS-PAGE.

## GTP-agarose pulldown assay
NRCMs were pretreated with 10 μM Na₂S₃ or 400 μM GSSG for 18 h, washed twice with DMEM, and then incubated under hypoxia or normoxia for an additional 18 h. 293 T cells expressing FLAG-Drp1 were cultured under hypoxia or normoxia for 18 h. NRCMs were washed with ice-cold PBS, lysed in GTP-binding buffer (50 mM Hepes pH7.4, 150 mM NaCl, 1.5 mM MgCl₂, 1 mM EGTA, 1% Triton X-100, 10% glycerol and protease inhibitor cocktail). The lysate was centrifuged (16,000 x g for 15 min at 4 °C), and an aliquot of the supernatant (100 μg protein) was incubated with 20 μl of GTP-agarose beads (equilibrated in GTP-binding buffer) for 45 min at room temperature. The beads were centrifuged (1000 x g for 1 min at 4 °C) and washed twice with GTP-binding buffer. The GTP-bound proteins were eluted with 2x Laemmli buffer containing dithiothreitol (DTT) and subjected to SDS-PAGE. Long-time and high-dose incubation of supersulfide donors into cardiomyocytes shows cytotoxicity[64]. Because of high reactivity, the cardioprotective range of Na₂S₄ is narrower than that of Na₂S₃. So, we used Na₂S₃ for cell-based experiment.

## Live cell imaging
NRCMs or hiPS-CMs were seeded at 2.0 or 1.5 x 10⁵ cells/ml on a matrigel-coated triple-well glass bottom dish (Iwaki). At 3 days after plating, cells were preincubated with 400 μM GSSG or GSH for 1 h and then cultured under hypoxia. For mitochondrial morphology, cells were incubated with 100 nM MitoBright Green LT for 30 min at 37 °C. Cells were washed with HBSS three times and imaged by BZ-X710 fluorescent microscope (Keyence). Mitochondria were traced and skeletonized to measure mitochondrial fragment length using ImageJ. The average mitochondrial fragment length of each cell was quantified and the mitochondrial morphology of each cell was classified into three groups (vesicle, intermediate and tubule) according to average mitochondrial fragment length (Vesicle, <4.7 μm; Intermediate, 4.7 – 6.4 μm; Tubule, > 6.4 μm)[6]. For mitochondrial membrane potential, cells were incubated with 2 μM JC-1 for 30 min at 37 °C, and the ratio of green to red intensity of JC-1 was quantified. For intracellular supersulfides[37] and hydrogen sulfide[35], cells were incubated with 10 μM SSip-1 in HBSS including 0.1% BSA and 0.04% Pluronic F-127 or 2.5 μM SF7-AM in HBSS for 30 min at 37 °C. The SSip-1 intensity was normalized by the maximum intensity induced by the treatment with 100 μM Na₂S₄. We used Na₂S₄ because of its high and fast reactivity compared with Na₂S₃. For simultaneous imaging of supersulfides and hydrogen sulfide, cells were incubated with 1 μM QS10[34] and 2.5 μM SF7-AM in HBSS including 0.04% Pluronic F-127 for 30 min at 37 °C. Cells were then washed five times with HBSS and images were taken every 30 sec for 15 min. For QS10 analysis, CH1 (Ex: 545 nm, Em: 605 nm) and CH2 (Ex: 545 nm, Em: 700 nm) intensities were measured and pseudo-color CH1/CH2 ratio images were generated using ImageJ for QS10 analysis. Intensity (Ex: 470 nm, Em: 525 nm) was measured for SF7-AM analysis.

## Proximity ligation assay (PLA)
Proximity ligation assay was conducted using Duolink PLA Fluorescence (Sigma Aldrich) according to the manufacturer's instructions. NRCMs were seeded at 2.0 × 10⁵ cells/ml on a matrigel-coated triple-well glass bottom dish. After fixing and blocking, NRCMs were incubated with rabbit anti-Drp1 (Santa Cruz) and mouse anti-s-glutathione (Abcam) antibodies. NRCMs were counterstained with DAPI and acti-nin or phalloidin. Images were captured using a BZ-X710 fluorescence microscope, and the number of PLA dots in cardiomyocytes was quantified.

## Measurement of OCR
Oxygen consumption rate (OCR) was assessed using an XFp Extra-cellular Flux Analyzer (Seahorse Bioscience, North Billerica, MA, USA). NRCMs were seeded on the plates with a density of 1.5 × 10⁴ cells/well in DMEM. The cells were treated with or without GSSG (400 μM) for 24 h. 30 min after addition of the compound, cells were incubated under 1% CSS or 1% O₂ conditions. Prior to analysis, cells were incubated for 1 h in XF Base Medium supplemented with 5 mM glucose, 1 mM pyruvate, and 2 mM glutamine. Cellular bioenergetics were measured by automatic injections of 10 μM oligomycin, 2 μM carbonyl cyanide-p-(trifluoromethoxy)phenylhydrazone (FCCP), 10 μM rote-none and 10 μM antimycin A, in order. After measurements of OCR, cells were fixed in 4% paraformaldehyde and washed twice with PBS. After stained with DAPI for 1 hour, cell images were captured, and the number of cell nuclei was counted using BZ-X800 microscope (Keyence). All values for OCR were normalized to the number of cells present in each well. OCR was measured from frozen heart using RIFS the respirometry protocol[69]. Briefly, frozen tissues were homogenized in MAS buffer (70 mM sucrose, 220 mM mannitol, 5 mM KH₂PO₄, 5 mM MgCl₂, 1 mM EGTA, 2 mM HEPES pH 7.4). Homogenates were centrifuged at 1000 x g for 10 min at 4 °C; then, the supernatant was collected. Three μg of each heart homogenates were suspended in 20 μL of MAS buffer and loaded into Seahorse XFp miniplates. The loaded plate was incubated with additional 130 μL of MAS buffer containing cytochrome c (10 μg/ml, final concentration). Substrate injection was as follows: 1 mM NADH at port A; 2.5 μM rotenone + 2.5 μM antimycin A at port B; 0.5 mM TMPD + 1 mM ascorbic acid at port C; and 50 mM azide at port D. Changes in OCR after addition of complex I substrate NADH was quantify for mitochondrial bioenergetics.

## Cardiomyocyte contraction

NRCMs or iPS-CMs were seeded at 4.0 or $3.0 \times 10^5$ cells/ml on a matrigel-coated glass bottom dish (Iwaki). At 3 days after plating, cells were preincubated with 400 μM GSSG for 1 h, and then cultured under hypoxia for 18 h. The medium was changed to phenol-red free DMEM, and cells were monitored through an inverted microscope IX-73 (Olympus) equipped with a 40x objective lens (LUCPlanFLN, Olympus) and a high-speed camera (WRAYCAM-VEX230M, Wraymer). Cells were electrically stimulated at 4 V and 1 Hz pacing frequency with a pulse width of 10 msec for 3 min, and then cell motion was recorded with imaging frame rates of 60 fps. The speed of contraction was measured using the ImageJ plugin MUSCLEMOTION[70].

## Statistical analysis

G*Power3.1.9.2 software was used to calculate the sample size for each group. The results are shown as means ± s.e.m. All experiments were repeated at least three times. Statistical comparisons were made with a two-tailed Student's t-test (for two groups), one-way analysis of variance (ANOVA) followed by Tukey's post-hoc test (for three and more groups) and two-way ANOVA followed by Sidak's post-hoc test (for time-course samples). Values of $P < 0.05$ were considered to be statistically significant.

"Generation of Drp1$^{C644S}$ knock-in mouse", "Purification of Drp1 protein for mass spectrometry", "Proteomic analysis of cysteine residues in Drp1", "MD simulation and computational analysis" and "Preparation of cigarette sidestream smoke (CSS)" sections were described in Supplementary methods.

## Reporting summary

Further information on research design is available in the Nature Portfolio Reporting Summary linked to this article.

## Data availability

All data generated and analyzed in this study are available in this article, its supplementary information files and from the corresponding author upon request. The mass spectrometry proteomics data generated in this study have been deposited in the ProteomeXchange Consortium via the PRIDE partner repository under accession code PXD053562. Source data are provided with this paper.

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

## Acknowledgements

We thank Prof. Ming Xian (Brown University) for the kind donation of chemical probes, and Ms. Hiromi Ishihara (National Institute for Physiological Sciences) for electron microscope analysis. This work was supported by JST CREST Grant Number JPMJCR2024 (20348438 to M.N. T.A. and A.N.), JSPS KAKENHI (24K02869 to A.N., 22H02772 and 22K19395 to M.N., 23K28237 to Yo.K., and 18H05277 and 22K19397 to T.A.), Grant-in-Aid for Scientific Research on Innovative Areas(A) "Sulfur biology" (21H05269 to M.N., 21H05263 to T.A., and 21H05258 to T.A. and M.N.) and International Leading Research (23K20040 to T.A.) from the Ministry of Education, Culture, Sports, Science and Technology of Japan, Joint Research of the Exploratory Research Center on Life and Living Systems (ExCELLS) (ExCELLS program No,23EX601), AMED (JP15km0908001), the Sumitomo foundation grant for basic science research project (to A.N.), Naito Foundation (to M.N.) and Smoking Research Foundation (to M.N.).

## Author contributions

M.N. and A.N. designed the research; A.N., X.T., and Yu.K. performed the experiment; K.H. and Y.S. performed bioinformatics; S.O. and T.A.

performed mass spectrometry analysis; A.N., S.O., K.U., Yu.K., Y.U. and T.A. analyzed and interpreted data; K.U., M.S., M.H., Y.I., Yo.K., K.K., Ya.K. and Y.U. contributed new reagents/analytic tools; A.N. and M.N. wrote the original manuscript. All authors contributed to the review and editing of the manuscript.

## Competing interests

The authors declare no competing interests.

## Additional information

[1]National Institute for Physiological Sciences, National Institutes of Natural Sciences (NINS), Okazaki, Japan. [2]Exploratory Research Center on Life and Living Systems, NINS, Okazaki, Japan. [3]SOKENDAI (The Graduate University for Advanced Studies), Okazaki, Japan. [4]Graduate School of Medicine, Tohoku University, Sendai, Japan. [5]Center for Computational Sciences, University of Tsukuba, Tsukuba, Japan. [6]Tokyo Metropolitan Institute for Geriatrics and Gerontology, Tokyo, Japan. [7]Graduate School of Pharmaceutical Sciences, Kyushu University, Fukuoka, Japan. [8]Graduate Division of Nutritional and Environmental Sciences, University of Shizuoka, Shizuoka, Japan. [9]Division of Pharmacology, National Institute of Health Sciences (NIHS), Kanagawa, Japan. [10]Graduate School of Pharmaceutical Sciences, The University of Tokyo, Tokyo, Japan. [11]Graduate School of Medicine, The University of Tokyo, Tokyo, Japan. ✉e-mail: nishida@phar.kyushu-u.ac.jp

