## [Transparent Peer Review file · Nature Communications]

Polysulfur-based bulking of dynamin-related protein 1 prevents ischemic sulfide catabolism and heart failure in mice

Corresponding Author: Professor Motohiro Nishida

Version 0:

Reviewer comments:

Reviewer #1

(Remarks to the Author)

Akiyuki Nishimura et al. reported GSSG, but not reduced GSH, prevents ischemic supersulfide catabolism-associated heart failure in mice by electrophilic modification of DRP1. Furthermore, they revealed a functional interaction between Cys644 and a critical phosphorylation site Ser637, through Glu640 and bulky modification at Cys644 via polysulfidation or S-glutathionylation reduced Drp1 activity by disrupting Ser637-Glu640-Cys644 interaction, which suggested a novel therapeutic potential of polysulfur-based Cys bulking on DRP1 for ischemic heart diseases. This study is well-organized and the methodology is sound. However, some concerns must be further addressed:

1. The study is basically in vitro study, the authors should supplement with in vivo models, such as genetically knockout or overexpressed DRP1 mice or genetically DRP1-mutated mice.
2. The author did not explain how they chose DRP1.
3. In Fig. 2, the author should add electron microscope morphology to intensify mitochondrial fission.
4. Seahorse assays should be supplemented for the mitochondrial function.
5. In Fig. 5, please examine the mitochondrial function and morphology.
6. The author detected the glutathionylation of DRP1 with GSH IP followed by DRP1 IB, is this method reliable? Please add some references.

Reviewer #2

(Remarks to the Author)

The manuscript by Akiyuki Nishimura et al. attempted to claim the protective effects of oxidized form of the glutathione (GSSG) against hypoxia induced cardiac dysfunction, which is somewhat interesting and novel. Authors demonstrated GSSG (oxidized glutathione), but not reduced GSH, prevents ischemic supersulfide catabolism-associated heart failure in mice by electrophilic modification of dynamin-related protein (Drp1). Authors suggesting electrophilic part of GSSG improves supersulfide catabolism-related mitochondrial dysfunction through Drp1 S-glutathionylation, protecting cardiomyocytes, and released GSH would also protect the heart from oxidative stress. While the study is somewhat interesting, not much of innovative, as it has been demonstrated the S-glutathionylation (often times, it is reversible post-translational modification) is protective. Several studies from Zweier's group have reported on these mechanisms. Although the claim is innovative that GSSG is protective against ischemic insult, the evidence to prove this is still insufficient. While increased glutathionylation is a hallmark for reductive stress. However, authors discussed their findings in relevance towards oxidative stress arm.

Other major concerns:

Fig 1A, Drp1 immunoprecipitate demonstrated higher levels of S-glutathionylation in the hearts from voluntary exercise mice (Ex) in comparison with sedentary mice (Sed). Technically, it is not clear how the samples were processed for this assay. How did the in vitro oxidation of -GSH prevented? The methodology is not clear and this might be a rigor/reproducibility concern. As my search for N-Ethylmaleimide (NEM) in the main text as well as the supplemental section failed to pull any hit. Therefore, it is clear that none of the samples were processed using NEM or equivalent reagent that prevents in vitro oxidation of -GSH while sample preparation. These small, but important, methodological issues may hugely reflect on the results as well as rigor/reproducibility.

Authors may consider adding more references related to reductive stress, as their results could potentially reflect a wide range of redox spectrum. In particular, author claim that GSSG prevents hypoxia-induced myocardial dysfunction. If this is a fact, (hypoxia induces myocardial damage/dysfunction), it is likely that prolonged hypoxic condition may lead to a reductive stress condition. Given that several chronic diseases have been recently linked to a chronic reductive redox milieu.

Cigarette side stream smoke (CSS) also produced similar changes in supersulfide catabolism with hypoxic stress in cardiomyocytes (Line 172-173). Considering similar changes, it is not clear whether GSSG administration showed different recovery phases?

How did the authors define hypoxic and pseudo hypoxic stress based on their findings?

Figures – 1D&F,3C&D,4C- IBs with more samples should be done to show the rigor and reproducibility in the experiments.

Figure 3D, authors presented the data from n=3 independent experiments. However, the densitometry/fold change graph showing n=4 points. This needs to be clarified with presenting the replicate data sets in supplemental section.

Given the poor permeability of the GSH or GSSG into cells, there is a basic concern on standardizing the cells with these small molecular thiols. This is applicable to the data from Fig.5 where the animal were treated with GSH and GSSG. Does this affect the turnover rates of glutathione following the treatment? Suggested that using glutathione ester (esterified glutathione) would more efficient in replenishing the glutathione levels in cells/organs.

Either blocking glutathione synthesis or supplementing glutathione is time-sensitive. For example, treating the animals with glutathione inhibitors work efficiently if administered between 8 to 10 am of the day. Treating the animals with GSSG (in any dose) would trigger global oxidative response. Thus, it is important to assure that the treatment did not result in too much of oxidative stress. At least, measuring the lipidperoxidaiton products or total antioxidant levels would aid in determining the dose of GSH or GSSG to be used. Optimization is crucial.

They have performed MD simulation for 500ns. But the dynamics of VD region compared with the range of 400-500 ns trajectories. What is the rationale for using this range?

Represented the hydrogen bonding interactions in terms of % of occupancy. If examined the dynamic properties of VD region for start to end trajectories and provide the number of hydrogen bonding across the percent of occupancy would be appropriate.

Based on the studies concluding that therapeutic potential of polysulfur-based Cys bulking on Drp1 for ischemic heart disease utilizes the GSSG/GSH ratio, echocardiography, cell imaging, proteome, and computation analysis. It is suggested to profile other markers (ex: p53 and senescence-associated β -galactosidase) of mitochondrial hyperfission and myocardial senescence to supportive the hypothesis.

Reviewer #3

(Remarks to the Author)

General comments:

Nishimura et al present interesting data sets on the posttranslational modifications of dynamin-related protein 1 in cardiomyocytes and conclude therapeutic concepts for the treatment of heart failure. Overall the authors obtained various data sets (Hypoxia in neonatal rat cardiomyocytes, sedentary versus exercise mice, cigarette sidestream smoking, cardiac function after myocardial infarction etc.) and especially the molecular simulation of Drp1 is well performed. How the different data sets are functionally related, however, remains in part unclear. The study would have profited from concentrating on one disease mechanism and to study this one in depth.

One major problem related to the hypoxia associated experiments is the undefined experimental setting. There is a controversy in the literature whether the release of electrons in the ETC is increased or even decreased in hypoxia. Since the authors did not define neither how they induced hypoxia nor the extent of hypoxia (oxygen concentration?), it is difficult to judge, if the proposed mechanism (see Fig. 7D) is likely to occur. Moreover, it is important to note that hypoxia does not resemble ischemia, which is part of the myocardial infarction disease pathology.

The manuscript is difficult to read based on English language problems.

Specific comments

Please provide control blots to demonstrate equal protein loading for Figure 1A.

Please provide control blots to demonstrate equal input for Figure 1D, E and F.

The authors state that Fig. 1E demonstrates less GSSG-mediated S-glutathionylation of Drp1 C644W compared to Drp1 wt. This is difficult to see from the figure provided since the intensity of the lanes shown seem to be very much similar for Drp1 wt and Drp1 C644W.

Hypoxia and hypoxia/reoxygenation are not defined neither in the result section nor in the Material and method section. The authors need to describe precisely how much oxygen was used and how the hypoxic conditions were established.

Figure 2 demonstrates that addition of GSSG to neonatal rat cardiomyocytes affects mitochondrial morphology, mitochondrial membrane potential and the number of SA- gal positive cells. The conclusion made by the authors "These results suggest S-glutathionylation mediated Drp1 inactivation by GSSG" is not justified based on the data presented.

The cigarette sidestream smoke data presented in Fig. 4 seem to be somewhat unrelated to the other data presented. Can the authors present a more precise justification how the data set underscores the overall conclusions? Stroke volume and cardiac output were not affected by the administration of GSH or GSSG according to Suppl Table 1 in sharp contrast to FS and EF (Fig. 5D and E). Is there any explanation for this discrepancy? How did the GSSG and GSH administration after myocardial infarction affect other organs (Fig. 5)? Is the protective effect unique for the heart?

Reviewer #4

(Remarks to the Author)

This MS describes how S-glutathionylation of Cys644 in the dynamin-related protein 1 by GSSG prevents mitochondrial hyperfission post-myocardial infarction and thus protects against the development of heart failure in rodents. The authors employed recombinant Drp1 protein and demonstrated polysulfidation in Cys644 in Drp1, which became S-glutathionylated as a result of GSSG electrophilic reaction by mass spectrometry, and determined how GSSG protects against mitochondrial hyperfission in hypoxia, cigarette smoke, and I/R-induced myocardial dysfunctions. Finally, the authors employed molecular dynamics to measure how S-glutathionylation could restrain Drp1 conformation, leading to its inactivation.

While the study provided evidence of GSSG protection of myocardial function against various insults, there is a lack of evidence and data to support the proposed biochemical mechanism of S-glutathionylation of Cys644 alone protecting against mitochondrial hyperfission. There is strong evidence that Cys644 is regulated by redox activity, first by Cho et al (2009) *Science* 10.1126/science.1171091 which shows that Cys644 is subjected to S-nitrosylation in mitochondria isolated from the brain and that nitrosative agent induces S-nitrosylation in Cys644 but not in mutant expressing C644A. More recently, Kim et al (2019) *Cell report* 10.1016/j.celrep.2018.05.054 also shows that protein disulfide isomerase protects against mitochondrial ROS production and sulfenylation of Cys644 in Drp1. The current study of GSSG could well be related to the role of redox enzymes in modulating the redox homeostasis in cells rather than direct targeting of Drp1 per se.

Unless the authors can provide more mechanistic data, this study shows just a broad protective effect of GSSG (overnight incubation is considered as very long) rather than a direct effect on Drp1.

1. The authors provided tandem MS spectra of S-glutathionylation of Cys644 (Fig. 1) and polysulfidated Cys644 (Supplementary Fig. 2), but not the redox modifications of the other cysteines in Drp1. This is essential to providing evidence that GSSG specifically targets Cys644 or all cysteines in general. Based on Fig. 1E, mutation of C644W only reduces S-glutathionylation by 50%, suggesting that other cysteines are also targeted by GSSG. Similarly, the authors should show GSSG treatment of WT Drp1 as compared to 8CS(C644) mutant in the same blot (Fig. 1F).
2. The authors should also show the relative abundance of all redox modifications of Drp1 cysteines (-SH, SSH, S-glutathionylation, S-nitrosylation, etc.) without or with GSSG treatment. The relative abundance of S-glutathionylated Cys644 and other cysteines is important. Since Drp1 oligomerizes leading to mitochondrial fission, one would expect Cys644 to be highly S-glutathionylated (the same for polysulfidation) in order to prevent oligomerization and fission. If only 5% of Cys644 is S-glutathionylated, then one needs to re-assess whether this is sufficient to block Drp1 oligomerization to prevent hyperfission of mitochondria. The fact that Cys644 can also be S-nitrosylated makes it more critical to measure the contribution of S-nitrosylation versus S-glutathionylation. The authors should be able to measure this in their mass spec data and show abundances of -SH, SSH, S-glutathionylation, S-nitrosylation, etc. relative to a control peptide. I tried to look for the mass spec raw files, but they were not provided for review.
3. Fig. 3B compares the GTPase activity of Drp1 without or with sodium trisulfide or with GSSG. It is unclear what the rationale is for using sodium trisulfide. The more commonly used hydrogen sulfide donors are sodium hydrosulfide (NaHS) or sodium sulfide (NaS). In Fig. 3C, the technique for measuring polyS Drp1 is unclear.
4. Fig. 4A is the only image data showing a change in supersulfide and hydrogen sulfide levels, yet the authors claimed there was a change in sulfide species. Positive and negative controls are needed using either genetic mutants and/or treatment with hydrogen sulfide donors to confirm. A change in polysulfide and hydrogen sulfide ratios can hardly be called sulfide metabolism. If the authors believe that there is a change in metabolism, more evidence, such as a change in the flux of metabolites, would be required to support this. I would suggest the authors consider a different title for this study.
5. The MD simulations study did not really make sense unless there is stronger evidence of S-glutathionylation of Cys644, as addressed in my earlier points.
6. Fig. 7 shows GTPase assays of wild-type Drp1 versus mutants. Although tryptophan has a bulky side chain, it is not the same as S-glutathionylation, which is an adduct of two amino acids at Cys644. The authors should demonstrate GTPase activity using enzymes such as glutaredoxin to show restoration of GTPase activity. The authors should also use a mutant of C644A to show that the addition of GSSG has no effect on the GTPase activity. Drp1 oligomerizes in vitro, and if a significant proportion of Cys644 in Drp1 is S-glutathionylated (enough to prevent hyperfission of mitochondria), the authors should be able to show differences in oligomerization in vitro without or with GSSG and also glutaredoxin. This will also remove doubt that S-glutathionylation of Cys644 regulates mitochondrial fission.

Version 1:

Reviewer comments:

Reviewer #1

(Remarks to the Author)

The authors have answered all of my concerning. The manuscript is now suitable for publication.

Reviewer #2

(Remarks to the Author)

The authors revised the manuscript with additional information to address the critiques. However, it still fails to differentiate the distinct effects of GSSG versus GSH treatment during cardiac injury. While some technical critiques were partially addressed, the manuscript does not convincingly support the impact of real-time redox changes during ischemia-mediated super sulfidation. Despite GSSG being proposed as a therapeutic target, it is crucial to measure real-time changes in ROS, including superoxide, hydroxide, and hydroperoxide. Furthermore, the mechanistic connection of Cys644 S-glutathionylation to the observed effects remains unclear. Additionally, the translational potential of these findings and their generalizability to human conditions are not evident.

Under hypoxic conditions during ischemia, mitochondrial metabolism is significantly altered, yet the connection between GSSG treatment and mitochondrial bioenergetics in intact hearts is not addressed.

Reviewer #3

(Remarks to the Author)

I would like to thank the authors, who managed to answer all questions. There is one doubt remaining: I could not find the description for generating the d Drp1 C644S mice in the Material and Method section (and would like to apologize in case I oversaw the information).

Reviewer #4

(Remarks to the Author)

The authors did a good job of addressing the comments on mass spectrometry analysis. The new data shown in Fig. 1I provides an explanation of why mutagenesis of C644W only led to a 40% decrease in S-glutathionylation since S-glutathionylation also occurs in the other Drp1 cysteines. The authors further showed that S-glutathionylation preferably occurs in polysulfidated Cys644 (Fig. 1J). It is surprising that the S-glutathionylation of Cys470 is as prominent as that of Cys644. Because the new data shows that GSSG could also target Cys470 as effectively as Cys644, the authors should revisit their mass spectrometry data to determine if Cys470 also becomes polysulfidated when exposed to supersulfide donors and whether S-glutathionylation preferably occurs at Cys470 that is polysulfidated as observed in Cys644. It would also be necessary to test the GTPase activity of C470S or C470W (Fig. 3C and D). This would be an important control to rule out inhibition of Drp1 activity by S-glutathionylated Cys470, as proposed by the authors (lines 366-367). The authors also demonstrated that co-expression of Grx1 partially attenuated the inhibitory effect of GSSG on Drp1 activation under hypoxia. As a control, the authors should also test if co-expression of Grx1 in the Drp1 C644S mutant would affect Drp1 activity to exclude the possibility that S-glutathionylation of Cys470 contributes Drp1 inhibition.

Minor comments

1. In Fig. 3D, images of the C644S mutant showed more punctate structures than the C644S mutant treated with GSSG, suggesting that GSSG treatment reduces Drp1 punctates. Furthermore, the data is expressed as the percentage of cells containing Drp1 punctates (Y-axis). It would be more useful to determine the average number of Drp1 punctates per cell in the wild-type and C644S mutants under different treatment conditions.
2. Lines 176-177 stated that GSSG treatment suppressed the particle formation of GFP-Drp1 but not GTP-Drp1 C644S under hypoxia (Fig. 3D). Only images of GFP-Drp1 C644S under normoxia were shown Fig. 3D. It would be useful to show images and quantifications of punctate structures of GFP-Drp1 C644S under hypoxia without or with GSSG treatment.
3. For supersulfide donors, are both sodium trisulfide (Na₂S₃) and sodium tetrasulfide (Na₂S₄) used interchangeably in the study? In the current manuscript, some experiments used Na₂S₄ (Fig. 1J; live cell imaging line 606), but others used Na₂S₃ (Fig. 3). Please clarify.
4. In supplementary Table 7, please specify the mass of the iodoacetamide adduct and the HPE-IAM adduct in the footnote or in the supplementary materials and methods.

Version 2:

Reviewer comments:

Reviewer #2

(Remarks to the Author)

I approve the revised manuscript for publication. I have no further comments, as my previous reviews were thorough and addressed all necessary questions.

Reviewer #4

(Remarks to the Author)

The authors have addressed all my comments in this revision and provided strong mechanistic evidence to support a protective role of S-glutathionylation at Drp1 Cys644 against mitochondrial fragmentation.

Responses to the Reviewers

First of all, we truly appreciate your great efforts to review our manuscript thoughtfully. We have addressed the concerns raised by the reviewer by adding explanations to each of the queries and with the addition of new experiments. Our response to each point is presented below and we denote where modifications to the text in the manuscript have been made by yellow highlight.

Response to Reviewer 1.

Akiyuki Nishimura et al. reported GSSG, but not reduced GSH, prevents ischemic supersulfide catabolism-associated heart failure in mice by electrophilic modification of DRP1. Furthermore, they revealed a functional interaction between Cys644 and a critical phosphorylation site Ser637, through Glu640 and bulky modification at Cys644 via polysulfidation or S-glutathionylation reduced Drp1 activity by disrupting Ser637-Glu640-Cys644 interaction, which suggested a novel therapeutic potential of polysulfur-based Cys bulking on DRP1 for ischemic heart diseases. This study is well-organized and the methodology is sound. However, some concerns must be further addressed:

- 1) The study is basically in vitro study, the authors should supplement with in vivo models, such as genetically knockout or overexpressed DRP1 mice or genetically DRP1-mutated mice.

[Response]

Thank you for your important comment. To elucidate our concept that polysulfidation and glutathionylation of Drp1 Cys644 is important for ischemic tolerance of mouse heart using in vivo model, we generated and evaluated Drp1 C644S hetero knock-in mouse (Drp1^{CS/+}). Basal cardiac parameters are almost the same between Drp1^{+/+} and Drp1^{CS/+} (supplemental fig 9), whereas cardiac functions of Drp1^{CS/+} were more vulnerable after myocardial infarction (MI) compared with that of Drp1^{+/+} (Fig. 5C-F). Moreover, the cardioprotective effects of GSSG administration were not observed in MI-operated Drp1^{CS/+} mice (Fig. 5G-I). This *in vivo* evidence would support our concept. This is described in the Result section (line 268-288).

- 2) The author did not explain how they chose DRP1.

[Response]

Thank you for your comment. We previously reported that mitochondrial dynamics through Drp1 activity has a crucial role in the regulation of stress tolerance of the heart (Ref. 6). Moreover, Drp1-mediated mitochondrial quality control is important for exercise performance in various organs (Ref. 24-25). Drp1 activity is regulated by not only phosphorylation but also various redox-related post-translational modifications (Ref. 13, 18, 27-30). Because physiological exercise induces changes in glutathione redox state in various species including human and mice (Ref. 31-33), we speculated the possibility that glutathione-related posttranslational modification (S-glutathionylation) of Drp1 occurs

in exercise mice for the regulation of mitochondrial dynamics and cardiac homeostasis. This is described in the Result section (line 114-118).

3) In Fig. 2, the author should add electron microscope morphology to intensify mitochondrial fission.

[Response]

Thank you for your advice. We observed the electron microscope morphology of mitochondria in NRCMs. Hypoxic stress increased cristae-disrupted mitochondria, and GSSG treatment rescued mitochondrial morphology (Fig. 2B). This is described in the Result section (line 158-159).

4) Seahorse assays should be supplemented for the mitochondrial function.

[Response]

Thank you for your advice. We performed Seahorse assay to analyze the effect of GSSG treatment on mitochondrial dysfunction induced by hypoxia/reoxygenation and electrophilic stimulation. Oxygen consumption rate (OCR) was recovered by GSSG treatment (Fig. 2D and Supplementary Fig. 6F). This is described in the Result section (line 159-162 for hypoxia/reoxygenation and 218-219 for electrophilic stimulation).

5) In Fig. 5, please examine the mitochondrial function and morphology.

[Response]

According to reviewer's comment, we added the experiments to analyze mitochondrial morphology and function from mouse hearts. We performed an electron microscope analysis to observe mitochondrial morphology (fragmentation) (Fig. 4G). We measured complex I activity and mtDNA/nDNA ratio to evaluate mitochondrial functions (Fig. 4H and I and Supplementary Fig. 7D). These results suggest that GSSG administration improves mitochondrial morphology and function in MI-operated mouse hearts. This is described in the Result section (line 250-255).

6) The author detected the glutathionylation of DRP1 with GSH IP followed by DRP1 IB, is this method reliable? Please add some references.

[Response]

We apologize for the lack of information on the IP-based assay for protein glutathionylation. S-glutathionylation of several proteins including actin, myosin and FABP5 has previously been detected by immunoprecipitation assay using monoclonal s-glutathione antibody that we used in this study. We added some original and review papers (Ref. 51, 75) about immunoprecipitation-based S-glutathionylation. This is described in the Method section (line 577-578).

Response to Reviewer 2.

The manuscript by Akiyuki Nishimura et.al. attempted to claim the protective effects of oxidized form of the glutathione (GSSG) against hypoxia induced cardiac dysfunction, which is somewhat interesting and novel. Authors demonstrated GSSG (oxidized glutathione), but not reduced GSH, prevents ischemic supersulfide catabolism-associated heart failure in mice by electrophilic modification of dynamin-related protein (Drp1). Authors suggesting electrophilic part of GSSG improves supersulfide catabolism-related mitochondrial dysfunction through Drp1 S-glutathionylation, protecting cardiomyocytes, and released GSH would also protect the heart from oxidative stress. While the study is somewhat interesting, not much of innovative, as it has been demonstrated the S-glutathionylation (often times, it is reversible post-translational modification) is protective. Several studies from Zweier's group have reported on these mechanisms. Although the claim is innovative that GSSG is protective against ischemic insult, the evidence to prove this is still insufficient. While increased glutathionylation is a hallmark for reductive stress. However, authors discussed their findings in relevance towards oxidative stress arm.

[Response]

Thank you for your critical comment. As the reviewer mentioned, the protective roles of protein glutathionylation have been reported in several organs. To evaluate the potential cardioprotective role of Drp1 glutathionylation, we first identified that Cys644 is one of the most effective targets of GSSG-mediated glutathionylation rather than other cysteines in Drp1 using quantitative mass spectrometry (Fig. 1I) in this revised manuscript. We next generated Drp1 C644S hetero knock-in mouse (Drp1^{CS/+}). Basal cardiac parameters are almost the same between Drp1^{+/+} and Drp1^{CS/+} (supplementary Fig. 9), whereas cardiac functions of Drp1^{CS/+} were more vulnerable after myocardial infarction (MI) compared with that of Drp1^{+/+} (Fig. 5C-F). Moreover, the cardioprotective effects of GSSG administration were not observed in MI-operated Drp1^{CS/+} mice (Fig. 5G-I). These results indicate that Drp1 Cys644 modification such as polysulfidation and glutathionylation is important for ischemic tolerance of the hearts and Drp1 Cys644 is required for ischemic cardioprotection by GSSG. We also identified that NADH/NAD⁺ ratio, reductive stress marker, is increased by MI, and GSSG but not GSH administration decreases reductive stress (Fig. 4J).

Other major concerns:

- 1) Fig 1A, Drp1 immunoprecipitate demonstrated higher levels of S-glutathionylation in the hearts from voluntary exercise mice (Ex) in comparison with sedentary mice (Sed).

Technically, it is not clear how the samples were processed for this assay. How did the in vitro oxidation of the –GSH prevented? The methodology is not clear and this might be a rigor/reproducibility concern. As my search for N-Ethylmaleimide (NEM) in the main text as well as the supplemental section failed to pull any hit. Therefore, it is clear that none of the samples were processed using NEM or equivalent reagent that prevents in vitro

oxidation of –GSH while sample preparation. These small, but important, methodological issues may hugely reflect on the results as well as rigor/reproducibility.

[Response]

We apologize for the lack of information about the assay to detect protein S-glutathionylation. We used the immunoprecipitation assay using anti-glutathione antibody. S-glutathionylation of several proteins including actin, myosin and FABP5 has previously been detected by immunoprecipitation assay. We added some original and review papers (Ref. 51, 75) about immunoprecipitation-based S-glutathionylation in the Method section (line 577-578). Drp1 Cys644 that is a target residue for S-glutathionylation is highly polysulfidated in the basal state (Fig. 1G), and polysulfidated Cys644 was preferentially S-glutathionylated by GSSG compared with CysSH, forming not only CysS-SG but also CysS-SSG and CysS-SSSG (Fig. 1J). Since we have previously found that NEM cleaves the S-S structure of polysulfidated cysteine, and the hydroxyphenyl group of tyrosine stabilizes polysulfidated cysteine by inhibiting the cleavage of S-S structure by oxidants such as hydroxyl anion (Hamid HA, Redox Biol., 21, 101096, 2019), we did not use NEM in this experiment for fear of adverse effects on the glutathionylation of polysulfidated Cys644 and instead, used tyrosine to protect S-glutathionylation of polysulfidated Cys644 from oxidants. This is described in the Method section (line 580-581).

- 2) Authors may consider adding more references related to reductive stress, as their results could potentially reflect a wide range of redox spectrum. In particular, author claim that GSSG prevents hypoxia-induced myocardial dysfunction. If this is a fact, (hypoxia induces myocardial damage/dysfunction), it is likely that prolonged hypoxic condition may lead to a reductive stress condition. Given that several chronic diseases have been recently linked to a chronic reductive redox milieu.

[Response]

Thank you for your comments. Excessive antioxidants, especially NADH are known as a key molecule of reductive stress and are closely associated with various types of chronic disorders. In our MI mouse model, NADH/NAD⁺ ratio was increased 5 weeks after MI operation (Fig. 4J), indicating NADH reductive stress conditions. We renewed the paragraph about reductive stress in the Discussion section with additional references related to reductive stress and reductive stress-mediated cardiomyopathy (line 381-389).

- 3) Cigarette side stream smoke (CSS) also produced similar changes in supersulfide catabolism with hypoxic stress in cardiomyocytes (Line 172-173). Considering similar changes, it is not clear whether GSSG administration showed different recovery phases?

[Response]

Thank you for your comment. As the reviewer points out, electrophiles such as cigarette sidestream smoke and methylmercury as well as hypoxic stress induced supersulfide catabolism in cardiomyocytes (Supplementary Fig. 5A, 5B, 6A and 6B). In revised experiments, mitochondrial uncoupler FCCP

induced supersulfide catabolism in a concentration and time-dependent manner (Supplementary Fig. 5D-H). Although the detailed molecular mechanism is still unclear, the disruption of mitochondrial membrane potential induced by hypoxic or electrophilic stress would be a key step for supersulfide catabolism. Enhanced supersulfide catabolism triggers depolysulfidation of Drp1, leading to mitochondrial fission-associated myocardial dysfunction (Fig. 2 and Supplementary Fig. 6). GSSG administration protected cardiomyocyte from hypoxic and electrophilic stress by inhibiting Drp1 activation through S-glutathionylation (Fig. 2 and Supplementary Fig. 6). However, as the other reviewer is concerned, our *in vivo* experiment was limited to ischemic stress, and the link to environmental stress is tenuous. Therefore, the CSS experiment was moved to the supplemental materials (Supplementary Fig. 6).

4) How did the authors define hypoxic and pseudo hypoxic stress based on their findings?

[Response]

Thank you for your question. We defined hypoxic and pseudo hypoxic stress according to HIF-1 α (hypoxia-inducible factor-1 alpha) expression. It is widely recognized that HIF-1 α is upregulated under low oxygen conditions as an oxygen-dependent transcription factor. It has been reported that cigarette smoke exposure induces HIF-1 α expression (Ref. 38). And we also confirmed HIF-1 α expression in cardiomyocytes by CSS exposure (Supplementary Fig. 6C), indicating CSS-mediated pseudo hypoxic stress. This is described in the Result section (line 213-215).

5) Figures – 1D&F,3C&D,4C- IBs with more samples should be done to show the rigor and reproducibility in the experiments.

[Response]

Thank you for your comments. We loaded multiple samples in the same SDS-PAGE gel and performed IB simultaneously for the rigor and reproducibility of IB experiments. We cropped and showed only one experiment because of limited figure space. We showed uncropped and multi-sample IB data in Supplementary Fig. 11-13.

We

6) Figure 3D, authors presented the data from n=3 independent experiments. However, the densitometry/fold change graph showing n=4 points. This needs to be clarified with presenting the replicate data sets in supplemental section.

[Response]

We apologize for our mistake. We performed n=4 independent experiments and quantified. We showed uncropped and multi-sample IB data of previous Fig. 3D (new Fig. 3G) in Supplementary Fig 12.

7) Given the poor permeability of the GSH or GSSG into cells, there is a basic concern on standardizing the cells with these small molecular thiols. This is applicable to the data from

Fig.5 where the animal were treated with GSH and GSSG. Does this affect the turnover rates of glutathione following the treatment? Suggested that using glutathione ester (esterified glutathione) would more efficient in replenishing the glutathione levels in cells/organs.

[Response]

Thank you for your advice. According to the reviewer's suggestion, we additionally performed *in vivo* experiment using glutathione reduced ethyl ester (GEE) that is a membrane-permeable GSH derivative. GEE administration did not improve cardiac functions of MI-operated mice (Supplementary Fig. 8 and Supplementary Tables 3 and 4), supporting that the oxidative ability of glutathione is important for protective effects against chronic heart disease models. This is described in the Result section (line 245-248).

8) Either blocking glutathione synthesis or supplementing glutathione is time-sensitive. For example, treating the animals with glutathione inhibitors work efficiently if administered between 8 to 10 am of the day. Treating the animals with GSSG (in any dose) would trigger global oxidative response. Thus, it is important to assure that the treatment did not result in too much of oxidative stress. At least, measuring the lipidperoxidaiton products or total antioxidant levels would aid in determining the dose of GSH or GSSG to be used. Optimization is crucial.

[Response]

Thank you for your advice. To continuously deliver glutathione, we used an osmotic pump system. To check oxidative stress level of the hearts, hearts were isolated from mice administrated vehicle, GSSG, GSH or GEE for 1 week. In our optimized concentration, these types of glutathione did not affect lipid peroxidation and total antioxidant levels in the heart (Supplementary Fig. 7B and C). This is described in the Result section (line 248-250).

9) They have performed MD simulation for 500ns. But the dynamics of VD region compared with the range of 400-500 ns trajectories. What is the rationale for using this range?

[Response]

Considering the insights from the structural stability analysis, we have chosen the trajectory range of 400-500 ns for our MD analysis. Through this investigation, we observed that the root mean square deviation (RMSD) analysis of C α atoms for WT, Cys-SSH, CW, and GSH indicates the attainment of an equilibrium phase after the 400 ns-MD trajectories (Figure R1). Additionally, our analysis of the C α -RMSD profiles for individual helices (H1-7) in all systems has supported our confidence in the structural stability observed. These analyses collectively affirm the suitability of the 400-500 ns trajectory range for our MD investigations.

Figure R1 Alpha carbon-root mean square deviation ($C\alpha$ -RMSD) analysis of the stalk domain and individual helices (H1-7) for WT, Cys-SSH, CW, and GSH is plotted versus time over a 500 ns trajectory. The color of the line plot for individual helices corresponds to the color of H1-7 in the 3D structure.

10) Represented the hydrogen bonding interactions in terms of % of occupancy. If examined the dynamic properties of VD region for start to end trajectories and provide the number of hydrogen bonding across the percent of occupancy would be appropriate.

[Response]

The analysis of hydrogen bonding (H-bond) interactions between S637 and E640 across the 500 ns-MD trajectories for WT, Cys-SSH, CW, and GSH has been plotted against the time axis, in alignment with the suggestion provided by the reviewer (Figure R2). Notably, the H-bond within the WT system exhibits greater prominence and stability compared to the other systems. This observation is consistent with the findings presented in Figure 6 of the manuscript, wherein the H-bond analysis is elaborated.

Figure R2 The hydrogen bonding interactions analysis between S637 and E640 over the 500 ns-MD trajectories for WT, Cys-SSH, CW, and GSH.

11) Based on the studies concluding that therapeutic potential of polysulfur-based Cys bulking on Drp1 for ischemic heart disease utilizes the GSSG/GSH ratio, echocardiography, cell imaging, proteome, and computation analysis. It is suggested to profile other markers (ex: p53 and senescence-associated β -galactosidase) of mitochondrial hyperfission and myocardial senescence to supportive the hypothesis.

[Response]

We additionally evaluated mitochondrial morphology and function of *in vivo* mouse hearts by electron microscopy, complex I activity assay and mtDNA/nDNA ratio assay (Fig. 4G-I). p53 is a known marker of senescence cells. We also checked myocardial senescence after MI operation and GSSG administration by not only SA- β -gal staining (Fig. 4L) but also p53 staining (Supplementary Fig 7E). All these experiments supported that GSSG but not GSH improves mitochondrial quality and cardiac remodeling after MI. This is described in the Result section (line 250-261).

Response to Reviewer 3.

General comments:

Nishimura et al present interesting data sets on the posttranslational modifications of dynamin-related protein 1 in cardiomyocytes and conclude therapeutic concepts for the treatment of heart failure. Overall the authors obtained various data sets (Hypoxia in neonatal rat cardiomyocytes, sedentary versus exercise mice, cigarette sidestream smoking, cardiac function after myocardial infarction etc.) and especially the molecular simulation of Drp1 is well performed. How the different data sets are functionally related, however, remains in part unclear. The study would have profited from concentrating on one disease mechanism and to study this one in depth.

One major problem related to the hypoxia associated experiments is the undefined experimental setting. There is a controversy in the literature whether the release of electrons in the ETC is increased or even decreased in hypoxia. Since the authors did not define neither how they induced hypoxia nor the extent of hypoxia (oxygen concentration?), it is difficult to judge, if the proposed mechanism (see Fig. 7D) is likely to occur. Moreover, it is important to note that hypoxia does not resemble ischemia, which is part of the myocardial infarction

disease pathology.

The manuscript is difficult to read based on English language problems.

[Response]

Thank you for your comments and we apologize for the lack of information about hypoxia. For *in vitro* hypoxia experiment, O₂ concentration in the multigas incubator was adjusted to 1%. This O₂ concentration is controlled by the injection of nitrogen gas. This is described in the Method section (line 555-557). To more analyze whether depolarized (impaired) mitochondria promote the reduction of supersulfide to H₂S in our model, we simultaneously analyzed the dynamic distribution of supersulfide and H₂S. Uncoupler FCCP treatment promoted the reduction of supersulfide to H₂S in a dose and time-dependent manner (Supplementary Fig. 5). We suggest that mitochondrial depolarization by hypoxia and electrophiles also induce supersulfide catabolism by electron from mitochondria.

We also agree with the reviewer's concern that hypoxia does not resemble ischemia in some respects. In our first manuscript, the cardioprotective role of Drp1 Cys644 glutathionylation was analyzed by just *in vitro* hypoxia model. Therefore, it was unclear whether Drp1 Cys644 glutathionylation is involved in the cardioprotective effect of GSSG against MI model mice. In our revised manuscript, we generated and evaluated Drp1 C644S hetero knock-in mouse (Drp1^{CS/+}) to elucidate our concept that polysulfidation and glutathionylation of Drp1 Cys644 is important for ischemic tolerance of mouse heart *in vivo*. Basal cardiac parameters are almost the same between Drp1^{+/+} and Drp1^{CS/+} (Supplementary Fig. 9), whereas cardiac functions of Drp1^{CS/+} were more vulnerable after myocardial infarction (MI) compared with that of Drp1^{+/+} (Fig. 5C-F). Moreover, the cardioprotective effects of GSSG administration were not observed in MI-operated Drp1^{CS/+} mice (Fig. 5G-I). This *in vivo* evidence would support our concept.

Specific comments

1) Please provide control blots to demonstrate equal protein loading for Figure 1A.

[Response]

Thank you for your comment. We showed GAPDH as a control blot in Fig. 1A. Uncropped and multi-sample blots were shown in Supplementary Fig. 11-13.

2) Please provide control blots to demonstrate equal input for Figure 1D, E and F.

[Response]

We showed total Drp1 as a control blot in Fig. 1D and E. Additionally Fig. 1F was deleted in the revised manuscript.

3) The authors state that Fig. 1E demonstrates less GSSG-mediated S-glutathionylation of Drp1 C644W compared to Drp1 wt. This is difficult to see from the figure provided since the intensity of the lanes shown seem to be very much similar for Drp1 wt and Drp1 C644W.

[Response]

We apologize for displaying an unclear figure. We have replaced it with a clearer figure of the differences in change, and shown a quantitative result. Glutathionylation of Drp1 C644W was reduced to 60% of the wild-type (Fig. 1E). We also evaluated Drp1 glutathionylation by quantitative mass spectrometry. Among all 9 cysteine residues in Drp1, Cys644 and Cys470 were most effectively S-glutathionylated by GSSG, and about half of Cys644 was S-glutathionylated by GSSG treatment (Fig. 1I). This mass spectrometry result is consistent with the result of immunoblot (Fig. 1E). This is described in the Result section (line 136-140).

4) Hypoxia and hypoxia/reoxygenation are not defined neither in the result section nor in the Material and method section. The authors need to describe precisely how much oxygen was used and how the hypoxic conditions were established. Figure 2 demonstrates that addition of GSSG to neonatal rat cardiomyocytes affects mitochondrial morphology, mitochondrial membrane potential and the number of SA- β -gal positive cells. The conclusion made by the authors "These results suggest S-glutathionylation mediated Drp1 inactivation by GSSG" is not justified based on the data presented.

[Response]

We apologize for the lack of information about hypoxia condition and the unsuitable conclusion in Fig. 2. For *in vitro* hypoxia experiment, O₂ concentration in the multigas incubator was adjusted to 1%. This O₂ concentration is controlled by the injection of nitrogen gas. This is described in the Method section (line 555-557). The conclusion of Fig.2 results was replaced with "These results suggest mitochondrial and cardio protective roles of GSSG in hypoxic cardiomyocytes" according to our evidence. This is described in the Result section (line 163-164).

5) The cigarette sidestream smoke data presented in Fig. 4 seem to be somewhat unrelated to the other data presented. Can the authors present a more precise justification how the data set underscores the overall conclusions?

[Response]

Thank you for your comment. In this paper, we showed that hypoxia-mediated supersulfide catabolism leads to mitochondrial hyperfission-associated myocardial dysfunction. And we have previously reported that various environmental factors such as methylmercury and cigarette smoking lead cardiomyocyte dysfunction through abnormal mitochondrial hyperfission (Ref. 18, 26). Therefore, we predicted the importance of sulfur redox status on cardiac risk caused by not only hypoxia but also environmental factors. To assess the importance and generality of sulfur redox status for cardiac homeostasis, we examined whether sulfide catabolism and Drp1 redox modification are involved in cardiac dysfunction induced by cigarette sidestream smoke (CSS) and whether GSSG has a protective effect. The results suggest that the redox status of Drp1 Cys644 plays an important role in CSS-mediated cardiac dysfunction. However, as the reviewer is concerned, our *in vivo* experiments (Fig. 4 and 5) were limited to ischemic stress, and the link to environmental stress is tenuous. Therefore, the CSS experiment

was moved to the supplemental materials (Supplementary Fig. 6).

- 6) Stroke volume and cardiac output were not affected by the administration of GSH or GSSG according to Suppl Table 1 in sharp contrast to FS and EF (Fig. 5D and E). Is there any explanation for this discrepancy?

[Response]

Thank you for your important comment. Stroke volume (SV) and cardiac output (CO) are an important index of cardiac performance. Although the obvious reason why SV and CO are not reduced in our MI model mice, the number of days after MI may be an important factor. It has been reported that FS but not SV is significantly decreased 1 month after ischemia-reperfusion (I/R) of C57BL6 mice, and SV is reduced 4 months after I/R (*Sci Rep.* 7, 14701, 2017). Since we only analyzed cardiac functions up to 1 month after MI, longer-term observations have been necessary to assess changes in SV. Additionally, in this paper, we evaluated SV using 4-dimensional echocardiography. There are several methods to analyze SV including catheterization, pulsed Doppler flow meter, 2-dimensional echocardiography and cardiac MRI. More multifaceted analysis may be needed to accurately assess SV. Therefore, we removed SV and CO parameters in the revised manuscript.

- 7) How did the GSSG and GSH administration after myocardial infarction affect other organs (Fig. 5)? Is the protective effect unique for the heart?

[Response]

Thank you for your comment. Several papers have reported that myocardial infarction leads the dysfunctions of liver and kidney (Aging 13, 2982-3009, 2021). To investigate whether kidney and liver are damaged in our MI model mice, we analyzed plasma biomarkers for kidney and liver and kidney fibrosis. Only blood urea nitrogen (BUN) that is a kidney marker was increased after MI (Fig. 4M and supplementary Fig. 7), suggesting mild effect on other organs. GSSG but not GSH administration inhibited BUN upregulation in MI-operated mice (Fig. 4M). However, further analysis would be required to determine whether GSSG protects kidney function directly or indirectly by protecting cardiac function. This is described in the Result section (line 261-265).

Response to Reviewer 4.

This MS describes how S-glutathionylation of Cys644 in the dynamin-related protein 1 by GSSG prevents mitochondrial hyperfission post-myocardial infarction and thus protects against the development of heart failure in rodents. The authors employed recombinant Drp1 protein and demonstrated polysulfidation in Cys644 in Drp1, which became S-glutathionylated as a result of GSSG electrophilic reaction by mass spectrometry, and determined how GSSG protects against mitochondrial hyperfission in hypoxia, cigarette smoke, and I/R-induced myocardial dysfunctions. Finally, the authors employed molecular dynamics to measure how

S-glutathionylation could restrain Drp1 conformation, leading to its inactivation.

While the study provided evidence of GSSG protection of myocardial function against various insults, there is a lack of evidence and data to support the proposed biochemical mechanism of S-glutathionylation of Cys644 alone protecting against mitochondrial hyperfission. There is strong evidence that Cys644 is regulated by redox activity., first by Cho et al (2009) Science 10.1126/science.1171091 which shows that Cys644 is subjected to S-nitrosylation in mitochondria isolated from the brain and that nitrosative agent induces S-nitrosylation in Cys644 but not in mutant expressing C644A. More recently, Kim et al (2019) Cell report 10.1016/j.celrep.2018.05.054 also shows that protein disulfide isomerase protects against mitochondrial ROS production and sulfenylation of Cys644 in Drp1. The current study of GSSG could well be related to the role of redox enzymes in modulating the redox homeostasis in cells rather than direct targeting of Drp1 per se.

Unless the authors can provide more mechanistic data, this study shows just a broad protective effect of GSSG (overnight incubation is considered as very long) rather than a direct effect on Drp1.

[Response]

Thank you for your critical comments. We fully agree that in the initial manuscript, there was not enough data to show that Drp1 Cys644 glutathionylation is directly involved in the cardioprotective effect of GSSG. According to reviewer's comments and suggestions, we added *in vitro* and *in vivo* evidence about a principal role of Drp1 Cys644 glutathionylation by GSSG-induced cardioprotection in this revised manuscript. We analyzed the modification status of Drp1 Cys644 using mass spectrometry analysis. Drp1 Cys644 was highly polysulfidated in basal state (Fig. 1G), and most effectively glutathionylated by GSSG compared with other cysteines in Drp1 (Fig. 1I). GSSG inhibited hypoxia-induced Drp1 activation. This protective effect of GSSG was not observed in glutathionylation-less Drp1 C644S mutant (Fig. 3C). Moreover, Glutaredoxin 1 (Grx1) overexpression decreased the inhibitory effect of GSSG on hypoxia-mediated Drp1 activation and multimer formation (Fig. 3D and E). Additionally, we confirmed that not only 18 h but also just 30 min pretreatment of GSSG showed Drp1 inhibition (Supplementary Fig. 4B) supporting a direct effect of GSSG on Drp1. Next, the critical role of Drp1 Cys644 in GSSG-mediated cardioprotective effects against *in vivo* mouse myocardial infarction (MI) model was evaluated. For this purpose, we generated and analyzed Drp1 C644S hetero knock-in mouse (Drp1^{CS/+}). Basal cardiac parameters are almost the same between Drp1^{+/+} and Drp1^{CS/+} (Supplemental Fig. 9), whereas cardiac functions of Drp1^{CS/+} were more vulnerable after MI compared with that of Drp1^{+/+} (Fig. 5C-F). Moreover, the cardioprotective effects of GSSG administration were not observed in MI-operated Drp1^{CS/+} mice (Fig. 5G-I). These *in vitro* and *in vivo* evidence suggest the pivotal role of Drp1 glutathionylation on the cardioprotective effect of GSSG.

- 1) The authors provided tandem MS spectra of S-glutathionylation of Cys644 (Fig. 1) and polysulfidated Cys644 (Supplementary Fig. 2), but not the redox modifications of the other

cysteines in Drp1. This is essential to providing evidence that GSSG specifically targets Cys644 or all cysteines in general. Based on Fig. 1E, mutation of C644W only reduces S-glutathionylation by 50%, suggesting that other cysteines are also targeted by GSSG. Similarly, the authors should show GSSG treatment of WT Drp1 as compared to 8CS(C644) mutant in the same blot (Fig. 1F).

[Response]

Thank you for your comments. We analyzed S-glutathionylation of all cysteine residues in Drp1 by GSSG treatment using mass spectrometry. Cys644 and Cys470 were preferentially glutathionylated by GSSG compared with other cysteines, and about 50% of these cysteines were glutathionylated (Fig. 1I). This mass spectrometry result is consistent with the finding that S-glutathionylation of Drp1 C644W mutant is reduced by approximately 40% compared to Drp1 WT (Fig. 1E). These results suggest that Cys644 is one of the most preferential target cysteine in Drp1 by GSSG. This is described in the Result section (line 136-140).

2) The authors should also show the relative abundance of all redox modifications of Drp1 cysteines (-SH, SSH, S-glutathionylation, S-nitrosylation, etc.) without or with GSSG treatment. The relative abundance of S-glutathionylated Cys644 and other cysteines is important. Since Drp1 oligomerizes leading to mitochondrial fission, one would expect Cys644 to be highly S-glutathionylated (the same for polysulfidation) in order to prevent oligomerization and fission. If only 5% of Cys644 is S-glutathionylated, then one needs to re-assess whether this is sufficient to block Drp1 oligomerization to prevent hyperfission of mitochondria. The fact that Cys644 can also be S-nitrosylated makes it more critical to measure the contribution of S-nitrosylation versus S-glutathionylation. The authors should be able to measure this in their mass spec data and show abundances of -SH, SSH, S-glutathionylation, S-nitrosylation, etc. relative to a control peptide. I tried to look for the mass spec raw files, but they were not provided for review.

[Response]

Thank you for your comments, and we apologize for the lack of information about mass spec raw data. The mass spectrometry proteomics data have been deposited to the ProteomeXchange Consortium via the PRIDE partner repository with the dataset identifier PXD053562. According to reviewer's suggestion, we analyzed the abundance of modification status of Drp1 without or with GSSG. The modification status of Drp1 Cys644 was directly measured by mass spectrometry analysis. In basal state, nearly 40% of Cys644 was polysulfidated (Fig. 1G). Considering previous reports that polysulfidated cysteine is unstable (Hamid HA, Redox Biol., 21, 101096, 2019), it would be expected that the population of polysulfidation at Cys644 is even higher prior to purification. This result would match our previous result that Cys644 polysulfidation negatively regulates Drp1 activity (Ref. 13, 18). On the other hand, S-glutathionylation and S-nitrosylation of Cys644 in basal state were not detected under our methodological condition. After GSSG treatment, approximately 50% of Cys644 was S-glutathionylated

(Fig. 1I). Additionally, polysulfidated Cys644 was preferentially S-glutathionylated by GSSG compared with CysSH (Fig. 1J). Although we were unable to detect S-nitrosylation in our detection system, we do not intend to deny previous reports about S-nitrosylation and oxidation of Cys644. Previous reports have been able to assess S-nitrosylation and oxidation of endogenous Drp1, albeit indirectly by using a biotin switch assay. In contrast, we used Drp1 overexpressed in HeLa cells to detect modification status directly by mass spectrometry. We attempted to purify endogenous Drp1 from hearts for mass spectrometry analysis, but the expression of Drp1 was too low to be successful. It is quite possible that the modification state differs between endogenous and exogenous Drp1. Additionally, S-nitrosylation and oxidation have been identified in the brain. Organ and cell-type may be important for variation of modification status of Drp1 Cys644. Considering our other experiments, we think that polysulfidation and S-glutathionylation of Cys644 are important in regulating Drp1 activity in cardiomyocytes. This is described in the Result (line 132-145) and Discussion (line 430-442) sections.

3) Fig. 3B compares the GTPase activity of Drp1 without or with sodium trisulfide or with GSSG. It is unclear what the rationale is for using sodium trisulfide. The more commonly used hydrogen sulfide donors are sodium hydrosulfide (NaHS) or sodium sulfide (NaS). In Fig. 3C, the technique for measuring polyS Drp1 is unclear.

[Response]

Thank you for your comments. According to reviewer's suggestion, we tested the effect of Na₂S pretreatment. Na₂S pretreatment and washout did not inhibit hypoxia-induced Drp1 activation as well as Na₂S₃ (Supplementary Fig. 4A). This is described in the Result section (line 167-170).

4) Fig. 4A is the only image data showing a change in supersulfide and hydrogen sulfide levels, yet the authors claimed there was a change in sulfide species. Positive and negative controls are needed using either genetic mutants and/or treatment with hydrogen sulfide donors to confirm. A change in polysulfide and hydrogen sulfide ratios can hardly be called sulfide metabolism. If the authors believe that there is a change in metabolism, more evidence, such as a change in the flux of metabolites, would be required to support this. I would suggest the authors consider a different title for this study.

[Response]

Thank you for your critical comments. As far as the reviewer is concerned, we analyzed the change in supersulfide and H₂S by imaging experiments. The selectivity of supersulfide probes (QS10, SSip-1) and H₂S probe (SF7-AM) have been reported in their original papers (*Angew Chem Int Ed Engl.* 57, 9346-9350, 2018 for QS10, *Chem Commun.* 53, 1064-1067, 2017 for SSip-1, *PNAS* 110, 7131-7135, 2013 for SF7-AM), and we also confirmed that QS10 but not SF7-AM reacts with Na₂S₂ (Supplementary Fig. 5C). Moreover, simultaneous imaging of supersulfide and H₂S showed that mitochondrial uncoupler FCCP decreased supersulfide and increased H₂S in concentration and time-dependent manner (Supplementary Fig. 5D-H). These imaging results revealed a strong negative correlation between

supersulfide and H₂S, which suggests the metabolic relationship between supersulfide and H₂S depending on mitochondrial activity. Additionally, our sulfur metabolomic analysis previously showed that supersulfide such as CysSSH can receive electrons from mitochondrial ETC and is converted to HS/H₂S for proper mitochondrial bioenergetics (*Nat Commun.* 8, 1177, 2017). Supersulfide-mediated electron flow increases the efficiency of the electron transport system and energy production (*Redox Biol.* 60, 102624, 2023). Although the detailed molecular mechanism is still unclear, the disruption of mitochondrial membrane potential would alter the balance of electron acceptance in ETC by oxygen and supersulfide and abnormally enhance supersulfide catabolism. This is described in the Result (line 183-195) and Discussion section (line 389-397).

5) The MD simulations study did not really make sense unless there is stronger evidence of S-glutathionylation of Cys644, as addressed in my earlier points.

[Response]

Thank you for your comment. In this revised manuscript, we performed additional experiments about S-glutathionylation of Drp1 Cys644 using mass spectrometry analysis, biochemical analysis and *in vivo* analysis of Drp1^{CS/+} mice (Fig. 1, 3 and 5). All these results support that S-glutathionylation as well as polysulfidation negatively regulate Drp1 activity. Hence, MD simulation was used to study the relationship between Cys644 modification and activity.

6) Fig. 7 shows GTPase assays of wild-type Drp1 versus mutants. Although tryptophan has a bulky side chain, it is not the same as S-glutathionylation, which is an adduct of two amino acids at Cys644. The authors should demonstrate GTPase activity using enzymes such as glutaredoxin to show restoration of GTPase activity. The authors should also use a mutant of C644A to show that the addition of GSSG has no effect on the GTPase activity. Drp1 oligomerizes *in vitro*, and if a significant proportion of Cys644 in Drp1 is S-glutathionylated (enough to prevent hyperfission of mitochondria), the authors should be able to show differences in oligomerization *in vitro* without or with GSSG and also glutaredoxin. This will also remove doubt that S-glutathionylation of Cys644 regulates mitochondrial fission.

[Response]

Thank you for your comments. According to reviewer's suggestion, we tested the effect of overexpression of Grx1 on the GTP-binding activity of Drp1 under hypoxia. Grx1 overexpression partially restored Drp1 activity in the condition of hypoxia and GSSG treatment (Fig. 3E). We also confirmed that GSSG failed to inhibit the activation of Drp1 C644S, suggesting the important role of Cys644 on the inhibitory effect by GSSG (Fig. 3C). Drp1 forms punctate and gather to mitochondrial fission site. We confirmed the cellular distribution of GFP-Drp1. GSSG treatment inhibited the punctate structure of GTP-Drp1 WT but not C644S. Moreover, Grx1 overexpression partially restored the

inhibitory effect of GSSG (Fig. 3D). All these results suggest the involvement of S-glutathionylation of Cys644 in GSSG-mediated Drp1 inactivation. This is described in the Result section (line 172-182).

Responses to the Reviewers

First of all, we truly appreciate your great efforts to review our manuscript thoughtfully. We have addressed the concerns raised by the reviewer by adding explanations to each of the queries and with the addition of new experiments. Our response to each point is presented below and we denote where modifications to the text in the manuscript have been made by yellow highlight.

Response to Reviewer 1.

The authors have answered all of my concerning. The manuscript is now suitable for publication.

[Response]

We appreciate your kind review.

Response to Reviewer 2.

The authors revised the manuscript with additional information to address the critiques. However, it still fails to differentiate the distinct effects of GSSG versus GSH treatment during cardiac injury. While some technical critiques were partially addressed, the manuscript does not convincingly support the impact of real-time redox changes during ischemia-mediated super sulfidation. Despite GSSG being proposed as a therapeutic target, it is crucial to measure real-time changes in ROS, including superoxide, hydroxide, and hydroperoxide. Furthermore, the mechanistic connection of Cys644 S-glutathionylation to the observed effects remains unclear. Additionally, the translational potential of these findings and their generalizability to human conditions are not evident.

Under hypoxic conditions during ischemia, mitochondrial metabolism is significantly altered, yet the connection between GSSG treatment and mitochondrial bioenergetics in intact hearts is not addressed.

[Response]

We appreciate your great efforts to review our manuscript. Molecular mechanism explaining the differences between GSSG and GSH treatment is a major focus of this study. GSSG but not GSH administration improved mitochondrial hyperfission and bioenergetics after MI (Fig. 4G and H and Supplementary Fig. 13A), suggesting that the protection of mitochondrial quality is the primary target of GSSG for cardioprotection. To analyze the mechanistic connection between Drp1 Cys644 S-glutathionylation and GSSG-mediated cardioprotection, we generated Drp1 C644S knock-in mouse in this study, and elucidated that Drp1 Cys644 is critical for ischemic tolerance of cardiomyocytes and cardioprotective role of GSSG *in vivo* (Fig. 5). Identification of Drp1 Cys644 S-glutathionylation from heart would provide a more compelling mechanistic link between Cys644 S-glutathionylation and cardioprotection. We attempted to purify endogenous Drp1 from hearts for mass spectrometry analysis, but no antibodies were available to sufficiently immunoprecipitate endogenous Drp1 as a limitation of this study. Therefore, we quantitatively analyzed polysulfidation and S-

glutathionylation of Drp1 using mass spectrometry analysis of exogenously expressed Drp1. Nearly 40% of Cys644 of Drp1-FLAG from HeLa cells was polysulfidated (Fig. 1G). Reconstitution assay using depolysulfidated Drp1 from *E. coli* showed that supersulfide donor Na_2S_4 treatment highly increased polysulfidation at Cys644 compared with other cysteines, and this polysulfidated Cys644 was preferentially S-glutathionylated by GSSG (Fig. 1J and Supplementary Fig. 3). Additionally, cell-based assay using Drp1 C644S and Grx1 also supports the critical role of Cys644 S-glutathionylation on GSSG-mediated mitochondrial protection (Fig. 3). In this revised manuscript, we tested the effect of GSH treatment on mitochondrial quality and showed that GSH treatment fails to prevent hypoxia-induced Drp1 activation and mitochondrial fission (Supplementary Fig. 7). These pieces of evidence suggest that GSSG protects cardiomyocytes from hypoxic stress by maintaining mitochondrial quality through Drp1 S-glutathionylation rather than GSH-mediated antioxidant effect. It has been reported that GSSG administration slightly increased the GSH levels in plasma and some organs such as liver and lung (Ref. 48), and decreased lipid peroxidation production in liver (Ref. 49). This previous evidence would speculate that either GSH or GSSG administration protects organs through antioxidant effects. However, antioxidant effects of GSSG have been reported when used at high concentrations. For example, in the paper on antioxidant effect of GSSG, about 700 mg/kg of GSSG was administered to mouse (Ref. 49). On the other hand, we used a lower dose (30 mg/kg/day) of GSSG or GSH in this study. In this revised manuscript, we compared antioxidant effect of GSSG and GSH administration in our condition using ischemia-reperfusion (IR) model, and found that GSH but not GSSG administration suppresses acute oxidative stress of IR model heart (Supplementary Fig. S11D and E). This result suggests that the antioxidant effect of glutathione is not sufficient for cardio-protective effects after MI, and GSSG protects the heart with mitochondrial protective effects rather than antioxidant effects. This is described in the Discussion section (line 397-426).

In this work, we focused on mitochondrial quality control to evaluate the cardioprotective role of GSSG taking into account our previous and recent results, and we could not analyze detailed changes in redox dynamics including ROS in mice after GSSG administration. Glutathione (GSH/GSSG balance) system plays a central role in regulating redox homeostasis. Although GSSG treatment did not affect lipid peroxidation and antioxidant activity in the basal heart (Supplementary Fig. 11B-C), it will be important to evaluate the effect of GSSG administration on redox balance including ROS and sulfur metabolites in whole body in detail as we assess clinical potentials of GSSG in the future. This is described in the Discussion section (line 427-432).

We truly agree with Reviewer's concerns about the generality of our findings. We tested the effect of GSSG on human iPS-derived cardiomyocytes (hiPS-CMs) in this revised manuscript. GSSG treatment improved hypoxia-induced mitochondrial fission and reduction of mitochondrial bioenergetics and cardiac contractility of hiPS-CMs (Supplementary Fig. 5A-C). This is described in the Result section (line 170-174).

Respirometry is the gold standard for evaluating mitochondrial bioenergetics, but usually requires freshly isolated mitochondria from intact heart. As we separated heart sample for electron microscopy, tissue imaging and biochemical analysis including BN-PAGE-based complex I activity assay, we could not prepare freshly isolated mitochondria. Therefore, we evaluated mitochondrial bioenergetics using a novel respirometry method from frozen heart homogenate as previously reported (Ref. 70). Changes in OCR after the addition of complex I substrate NADH was decreased in MI heart sample and this impairment was recovered by GSSG but

not GSH treatment (Supplementary Fig. 13A). Consistent with BN-PAGE-based Complex I activity assay (Fig. 4H and Supplementary Fig. 13B), it is suggested that GSSG but not GSH improves mitochondrial bioenergetics. This is described in the Result (line 276) and Method (line 648-657) section.

Response to Reviewer 3.

I would like to thank the authors, who managed to answer all questions. There is one doubt remaining: I could not find the description for generating the d Drp1 C644S mice in the Material and Method section (and would like to apologize in case I oversaw the information).

[Response]

We appreciate your kind review and additional comment. We added method to generate Drp1 C644S mice in Supplementary Method section.

Response to Reviewer 4.

The authors did a good job of addressing the comments on mass spectrometry analysis. The new data shown in Fig. 1I provides an explanation of why mutagenesis of C644W only led to a 40% decrease in S-glutathionylation since S-glutathionylation also occurs in the other Drp1 cysteines. The authors further showed that S-glutathionylation preferably occurs in polysulfidated Cys644 (Fig. 1J). It is surprising that the S-glutathionylation of Cys470 is as prominent as that of Cys644. Because the new data shows that GSSG could also target Cys470 as effectively as Cys644, the authors should revisit their mass spectrometry data to determine if Cys470 also becomes polysulfidated when exposed to supersulfide donors and whether S-glutathionylation preferably occurs at Cys470 that is polysulfidated as observed in Cys644. It would also be necessary to test the GTPase activity of C470S or C470W (Fig. 3C and D). This would be an important control to rule out inhibition of Drp1 activity by S-glutathionylated Cys470, as proposed by the authors (lines 366-367). The authors also demonstrated that co-expression of Grx1 partially attenuated the inhibitory effect of GSSG on Drp1 activation under hypoxia. As a control, the authors should also test if co-expression of Grx1 in the Drp1 C644S mutant would affect Drp1 activity to exclude the possibility that S-glutathionylation of Cys470 contributes Drp1 inhibition.

[Response]

Thank you for your important suggestion. We completely agree to clear the role of the S-glutathionylation of Cys470 on Drp1 activity. Firstly, we reanalyzed mass spectrometry data about the modification of Cys470 and Cys644. High dose GSSG (3 mM) preferentially induced S-glutathionylation at Cys470 and Cys644 (Fig. 1I). In less-polysulfidated Drp1 purified from *E. coli*, Cys470-SH, Cys644-SH and other cysteines showed a similar and low efficiency for S-glutathionylation by a low dose (1 mM) of GSSG (Supplementary Fig. 3). However, Cys644 was preferentially polysulfidated by Na₂S₄, and this polysulfidated Cys644 was highly S-glutathionylated by GSSG compared with other cysteines including Cys470, suggesting

that polysulfidated Cys644 is a more preferred target for GSSG-mediated S-glutathionylation (Supplementary Fig. 3). This is described in the Result section (line 143-151). Second, we analyzed the activity of C470S mutant. We previously reported that the overexpression of Drp1 C644S but not C470S induces mitochondrial fragmentation (Ref. 18). Consistent with this, Drp1 C470S mutant showed low GTP-binding and multimer formation activity in basal condition, and hypoxia-induced activation of Drp1 C470S is drastically inhibited by GSSG treatment (Supplementary Fig. 8A and B). Moreover, co-expression of Grx1 did not alter the GTP-binding activity of Drp1 C644S with or without GSSG (Supplementary Fig. 8C). This is described in the Result section (line 193-199). These additional results would support that Cys644 but not Cys470 is important for GSSG-mediated inactivation of Drp1.

Minor comments

1. In Fig. 3D, images of the C644S mutant showed more punctate structures than the C644S mutant treated with GSSG, suggesting that GSSG treatment reduces Drp1 punctates. Furthermore, the data is expressed as the percentage of cells containing Drp1 punctates (Y-axis). It would be more useful to determine the average number of Drp1 punctates per cell in the wild-type and C644S mutants under different treatment conditions.

[Response]

Thank you for your advice. We reanalyzed imaging data and quantified the average number of Drp1 punctates per cell (Fig. 3D). Re-analyzed results showed a similar trend to the previous results. The image of Drp1 C644S without GSSG was changed to a more representative one.

2. Lines 176-177 stated that GSSG treatment suppressed the particle formation of GFP-Drp1 but not GTP-Drp1 C644S under hypoxia (Fig. 3D). Only images of GFP-Drp1 C644S under normoxia were shown Fig. 3D. It would be useful to show images and quantifications of punctate structures of GFP-Drp1 C644S under hypoxia without or with GSSG treatment.

[Response]

Thank you for your suggestion. We added data of imaging of punctate structures of GFP-Drp1 C644S under hypoxia without or with GSSG treatment (Fig. 3D). GSSG treatment did not affect the punctate structures of GFP-Drp1 C644S under hypoxia.

3. For supersulfide donors, are both sodium trisulfide (Na₂S₃) and sodium tetrasulfide (Na₂S₄) used interchangeably in the study? In the current manuscript, some experiments used Na₂S₄ (Fig. 1J; live cell imaging line 606), but others used Na₂S₃ (Fig. 3). Please clarify.

[Response]

Thank you for your comments. Na₂S₄ shows higher reactivity than Na₂S₃. So, we used Na₂S₄ for the recombinant protein experiment (Fig. 1J). However, long-time and high-dose incubation of supersulfide donors into cardiomyocytes shows cytotoxicity as previously reported (Akiyama et al, Redox Biol, 2022). Because of high reactivity, the Cardioprotective range of Na₂S₄ is narrower than that of Na₂S₃. So, we used Na₂S₃ for cell-based

experiment (Fig. 3B). This is described in the Method section (line 605-608).

For quantification of SSip-1 signal, Fmax signal was finally obtained by adding supersulfide donor. To rapidly and completely saturate SSip-1 signal, we used highly reactive Na_2S_4 rather than Na_2S_3 (Supplementary Fig 10A). This is described in the Method section (line 621-622).

4. In supplementary Table 7, please specify the mass of the iodoacetamide adduct and the HPE-IAM adduct in the footnote or in the supplementary materials and methods.

[Response]

Thank you for your comments. We added the information about the mass of the iodoacetamide adduct and the HPE-IAM adduct in the footnote of Supplementary Table 7.

Responses to the Reviewers

We truly appreciate your great efforts to review our manuscript thoughtfully.

Response to Reviewer 2.

I approve the revised manuscript for publication. I have no further comments, as my previous reviews were thorough and addressed all necessary questions.

[Response]

We appreciate your kind review.

Response to Reviewer 4.

The authors have addressed all my comments in this revision and provided strong mechanistic evidence to support a protective role of S-glutathionylation at Drp1 Cys644 against mitochondrial fragmentation.

[Response]

We appreciate your kind review.